# Unveiling the complexity of spatiotemporal soliton molecules in real time

Yuankai Guo [1,4], Wei Lin[1,4], Wenlong Wang[1], Runsen Zhang[1], Tao Liu[1], Yiqing Xu [2], Xiaoming Wei[1] ✉ & Zhongmin Yang[1,3] ✉

Observing the dynamics of 3D soliton molecules can hold great opportunities for unveiling the mechanism of molecular complexity and other nonlinear problems. In spite of this fantastic potential, real-time visualization of their dynamics occurring on femtosecond-to-picosecond time scales is still challenging, particularly when high-spatiotemporal-resolution and long-term observation are required. In this work, we observe the real-time speckle-resolved spectral-temporal dynamics of 3D soliton molecules for a long time interval using multispeckle spectral-temporal measurement technology. Diverse real-time dynamics of 3D soliton molecules are captured for the first time, including the speckle-resolved birth, spatiotemporal interaction, and internal vibration of 3D soliton molecules. Further studies show that nonlinear spatiotemporal coupling associated with a large average-chirp gradient over the speckled mode profile plays a significant role in these dynamics. These efforts may shed new light on decomposing the complexity of 3D soliton molecules, and create an analogy between 3D soliton molecules and chemical molecules.

In the field of molecular sciences, fundamental questions remain open, such as how atoms behave to change the chemical and physical properties of molecules, and how molecules interplay to affect the macro properties of matters[1,2]. While experimentally observing the real-time motion of atoms and molecules is still difficult[3–5], it can be studied by referencing the dynamics of three-dimensional (3D) soliton molecules in nonlinear optical systems, which may share many common characteristics with the dynamics of chemical molecules[6]; while one-dimensional (1D) soliton generated through nonlinear interaction in fiber optics has been intensively studied and correlated to chemical molecular dynamics[7–9]. In particular, the various physical variables in 3D nonlinear optical systems[10,11], e.g., the polarization, wavelength, dispersion (both chromatic and modal ones), nonlinearity, gain, and loss, provide additional degrees of freedom for manipulating 3D soliton molecules. The perturbation to the physical parameter space, in

addition, can create versatile dynamics of 3D soliton molecules through spatiotemporal coupling (STC)−increasing the possibility of making analogies to the behaviors of chemical molecules. As a result, observing the internal motion of soliton molecules as well as their interplay can be a powerful tool for tackling molecular problems that are not experimentally straightforward in chemistry.

The motions of 3D optical solitons and their bound states (i.e., soliton molecules) on femtosecond-to-picosecond time scales, however, usually start at an unpredictable time (i.e., largely non-repeatable), and the subsequent evolution may last for several milliseconds. Time-averaged studies of 3D soliton molecules[12,13] and real-time measurements of one-dimensional solitons[14–23] have been recently demonstrated. In the meantime, super-high-speed photography technologies, especially the compressed ultrafast photography (CUP)[24], have been successfully demonstrated, and interesting findings have

[1]School of Physics and Optoelectronics, State Key Laboratory of Luminescent Materials and Devices, Guangdong Engineering Technology Research and Development Center of Special Optical Fiber Materials and Devices, Guangdong Provincial Key Laboratory of Fiber Laser Materials and Applied Techniques, South China University of Technology, 381 Wushan Road, Guangzhou 510640, China. [2]Department of Physics, University of Auckland, Auckland 1010, New Zealand. [3]Research Institute of Future Technology, South China Normal University, Guangzhou, Guangdong 510006, China. [4]These authors contributed equally: Yuankai Guo, Wei Lin. ✉e-mail: xmwei@scut.edu.cn; yangzm@scut.edu.cn

been obtained in the field of soliton dynamics[25] and optical chaos[26]. However, real-time observing the dynamics of 3D soliton molecules by far is largely unmet, as it presses for a real-time measurement technology with a high spatiotemporal resolution, long recording length, and multiple measurement dimensions. In this work, we demonstrate speckle-resolved spectral-temporal dynamics of 3D soliton molecules in real time using a single-shot multispeckle spectral-temporal measurement technology. Versatile spatio-spectral-temporal dynamics of 3D soliton molecules are observed for the first time, and the underlying mechanism is also investigated. These results can pave the way to decomposing the complexity of 3D soliton molecules.

## Results

### Principle of 3D soliton molecule dynamics

Figure 1a conceptually illustrates the generation of 3D soliton molecules that can be utilized as a promising interdisciplinary platform for exploring the complexity of chemical molecules. Firstly, we establish a numerical simulation platform to predict the dynamics of 3D soliton molecules (Supplementary Note 1). In the numerical simulation, diverse dynamics can be generated, including the birth of 3D soliton molecules (Supplementary Fig. 1), their internal motion (Supplementary Fig. 2), and speckle-resolved evolutions (Supplementary Figs. 3–7). The numerical results indicate the complex behaviors of 3D soliton molecules in both the time and space domains. In the experiment, 3D optical solitons are generated through spatiotemporal mode-locking in a multimode oscillator[6], specifically, a spatiotemporal

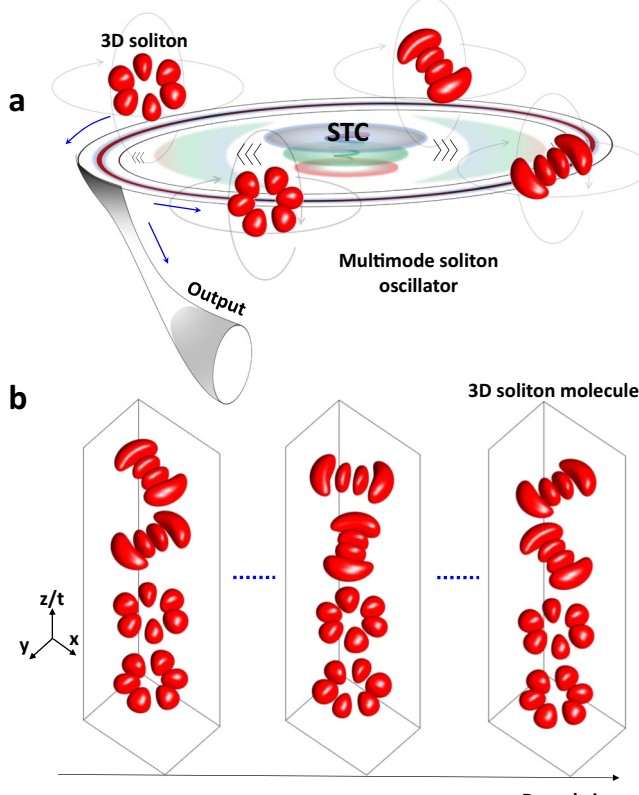

**Fig. 1 | Formation of 3D soliton molecules in a multimode soliton oscillator.**
**a** Conceptual diagram of 3D soliton molecules generated in a multimode soliton oscillator. Multiple 3D optical solitons can be generated in a spatiotemporal mode-locked fiber laser (numerical simulation is attached in Supplementary Note 1). The coexisted 3D optical solitons are bound together as 3D soliton molecules through spatiotemporal coupling (STC). The generated 3D soliton molecules then internally and externally evolve, giving rise to diverse spatial-temporal-spectral dynamics.
**b** Simulated evolution of 3D soliton molecules over multiple roundtrips. The 3D isosurface plot is utilized for better visualization.

mode-locked laser constructed with multimode fibers in this case (Supplementary Note 2). The spatial and spectral filtering effects produced by fiber coupling and bandpass filtering, respectively, are employed to facilitate the self-started mode-locking. The 3D nature of generated optical solitons can be identified by the complicated mode profile, as shown in Supplementary Fig. 8, wherein multiple speckle grains (SGs) resulted from the coherent interference of the 3D optical solitons are clearly presented. The spatiotemporal complexity of the 3D optical solitons can also be partially visualized from the pulse width variation across the speckled mode profile (right inset of Supplementary Figure 8). Please note that, here the operation of the multimode soliton oscillator is deliberately detuned from its optimal state— usually serving as an ideal platform for generating high-quality femtosecond 3D pulses[6], such that 3D soliton molecules can be excited through STC effect. Thanks to the spatiotemporal interplay between 3D optical solitons, 3D soliton molecules exhibit complex transient dynamics, as illustrated in Fig. 1b.

The output of the multimode soliton oscillator is launched to a real-time multispeckle spectral-temporal (MUST) measurement system[27]. In brief, the MUST measurement system has three single-mode fiber probes that can simultaneously extract the spectral-temporal information of three speckle grains in real time (Supplementary Note 3). Specifically, the optical time division multiplexing (OTDM) technology widely used in optical communication is adopted, and the signals from the three probes are temporally multiplexed (Supplementary Fig. 9). The temporally multiplexed signal is then split into two branches. One branch is directed to a high-speed photodiode for real-time observation in the time domain. The other is launched to a wavelength-to-time unit that performs dispersive Fourier transformation (DFT) in a double-pass scheme[28,29], enabling real-time spectroscopy. The signals from the two branches are simultaneously recorded by a multi-channel real-time oscilloscope.

### Dynamics of 3D dual-soliton molecules

Figure 2 presents the spectral-temporal dynamics after the formation of the 3D dual-soliton molecule in two different speckle grains, recorded by the MUST measurement system. As shown in Fig. 2a, b, the spatiotemporal structure of the 3D soliton molecule gives rise to the dense spectral interference fringes. Interestingly, the resulting spectral interference fringes differ between speckle grains (i.e., $SG_1$ and $SG_2$ in this case). In particular, the spectral interference fringes of $SG_1$ (Fig. 2c) exhibit additional intensity modulation in contrast to that of $SG_2$ (Fig. 2d), implying the spatiotemporal interplay between the 3D optical solitons. The 3D optical solitons slowly evolve in the $x$, $y$, $t$ and $z$ dimensions through nonlinear coupling, leading to various spectral-temporal landscapes in different speckle grains.

To reveal the corresponding temporal dynamics, we perform field autocorrelation (FAC, see Supplementary Note 4) calculations from the results of spectral evolutions, as shown in Fig. 2e, f. It manifests that there exists another drifting optical soliton in $SG_1$, as indicated in Fig. 2e, which is not bound with the dual-soliton molecule. The spatiotemporal interaction between the drifting soliton and the dual-soliton molecule results in the intensity modulation in $SG_1$. Here, the results highlight the importance of the FAC calculation as a complementary tool for unveiling the dynamic features in the time domain. It is also worth noting that, thanks to the real-time spectral measurement over a long period of time and corresponding FAC calculation, the birth of 3D soliton molecules and their slow motion after formation are completely observed (Supplementary Note 5).

To understand the underlying physics, we perform (3 + 1)-D numerical simulations. Figure 3a–c shows the temporal evolutions of the 3D dual-soliton molecule obtained in the numerical simulation, i.e., the evolutions in the complete mode area, $SG_1$ and $SG_2$, respectively (also see their spectral evolutions in Supplementary Fig. 20). Similar to that of Fig. 2, in addition to the 3D dual-soliton molecule, there exists a

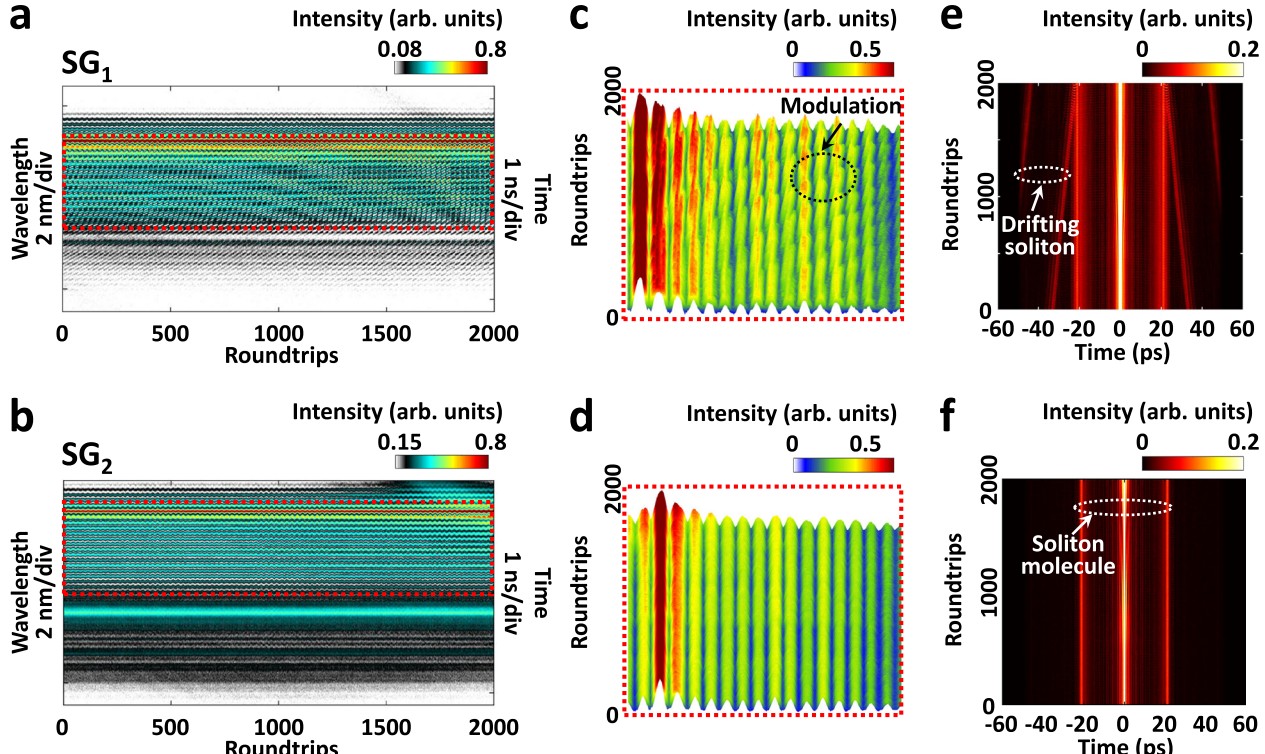

**Fig. 2 | Speckle-resolved spectral-temporal dynamics of 3D dual-soliton molecules. a, b** Spectral evolutions of the 3D dual-soliton molecule in two different speckle grains. **c, d** Close-ups of the spectral evolutions, as indicated by the red dotted squares in (**a**) and (**b**). Here, 3D plots are used for better visualization. **e, f** Corresponding field autocorrelation (FAC) evolutions of (**a**) and (**b**), respectively.

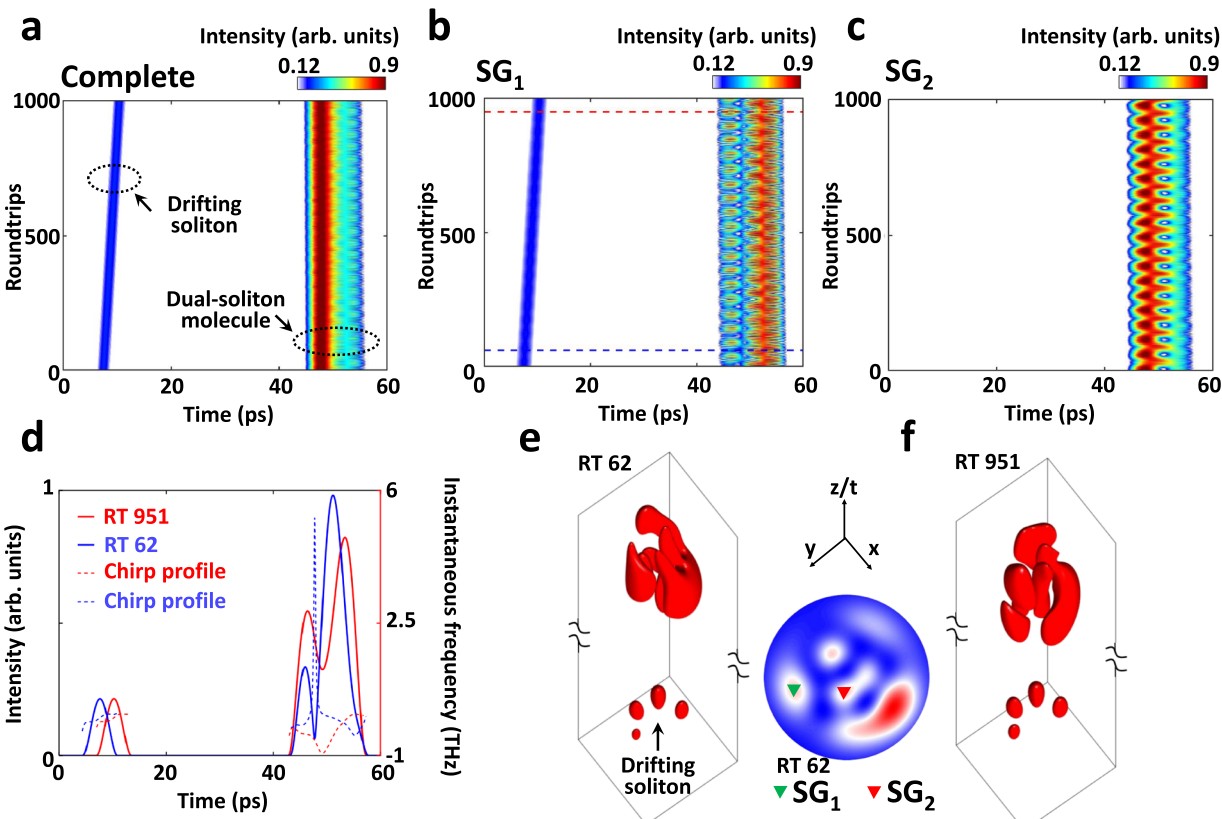

**Fig. 3 | Numerical simulation of 3D dual-soliton molecules. a** Temporal evolution of the 3D dual-soliton molecule in the complete mode area, i.e., the temporal signal averaged from all the speckle grains. **b, c** Temporal evolutions in two different speckle grains (i.e., SG$_1$ and SG$_2$). The corresponding spectral evolutions are provided in Supplementary Fig. 20. **d** Intensity and chirp profiles of roundtrips (RTs) 62 and 951 in SG$_1$, as indicated in **b**. **e, f** Isosurface plots of the 3D dual-soliton molecule associated with a drifting soliton in roundtrips 62 and 951. The averaged mode profile is also provided (middle), wherein the SG$_1$ and SG$_2$ are indicated by the green and red triangles, respectively.

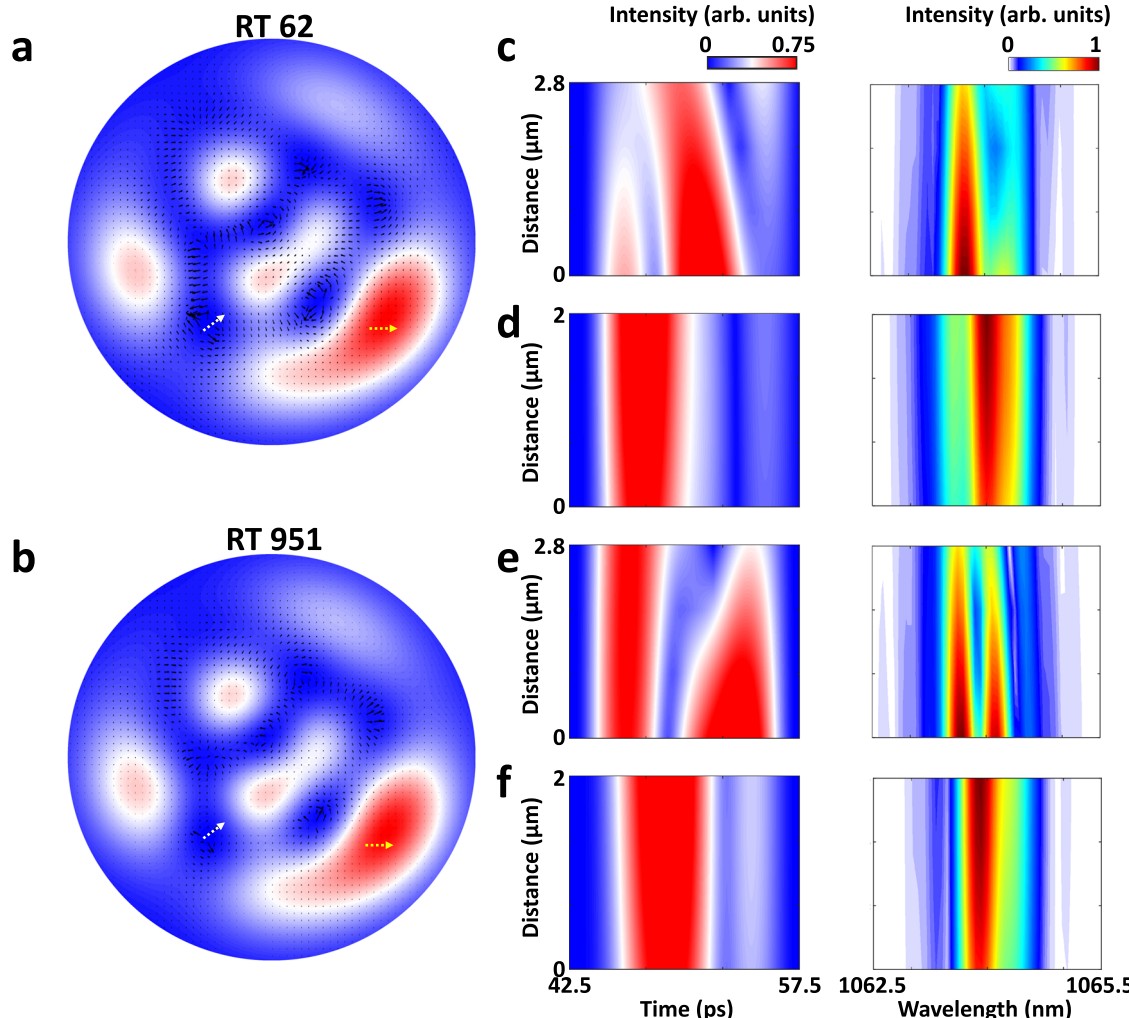

**Fig. 4 | Spatio-spectral-temporal properties of 3D dual-soliton molecules in the simulation. a, b** 2D gradient (black arrows) of the average chirp of the 3D soliton molecule across the speckled mode profile, in roundtrips 62 and 951, respectively. **c, d** Temporal (left) and spectral (right) evolutions along the white and yellow arrows indicated in (**a**). **e, f** Temporal (left) and spectral (right) evolutions along the white and yellow arrows indicated in (**b**).

drifting soliton that gradually approaches the dual-soliton molecule, as indicated in Fig. 3a. Especially, the temporal evolution in $SG_1$ (Fig. 3b) shows a landscape involving both the dual-soliton molecule and drifting soliton, while only the dual-soliton molecule is presented in $SG_2$ (Fig. 3c), which identifies the experimental observation (Fig. 2). Furthermore, Fig. 3c manifests that the temporal intensity profile of the 3D dual-soliton molecule periodically varies over roundtrips, and different evolution patterns present in the speckle grains. The same case exists in their spectral evolutions (Supplementary Fig. 20). Such periodically breathing phenomenon can be attributed to the spatio-temporal interaction between the solitons. The decomposed temporal evolutions of different transverse modes also exhibit diverse temporal evolution landscapes (Supplementary Fig. 21).

The chirp of the drifting soliton and dual-soliton molecule, in addition, exhibits different profiles, as shown in Fig. 3d. The chirp profile of the drifting soliton is maintained over roundtrips, while it drastically changes for the dual-soliton molecule. This further suggests that spatiotemporal interaction occurs between the solitons of the dual-soliton molecule as they co-propagate over roundtrips, imparting physical changes to each other through intramodal and intermodal nonlinear effects. Their 3D profiles also indicate the spatiotemporal change of the dual-soliton molecule over roundtrips (Fig. 3e, f).

To probe the origin of these spectral-temporal dynamics, the 2D gradient of the average chirp of the 3D soliton molecule is calculated for the speckled mode profile (see Methods), as shown in Fig. 4a, b (i.e., the black arrows). Here, the direction of the black arrow indicates the direction of the average-chirp gradient, and its length represents the increasing rate of the average chirp. The simulation results imply that, in the area with a large average-chirp gradient, e.g., the white arrow of Fig. 4a, the temporal and spectral profiles significantly vary along the gradient direction (Fig. 4c). In contrast, they remain relatively stable in the area with a small average-chirp gradient (Fig. 4d). Notably, the temporal and spectral evolutions in the area with a large average-chirp gradient dramatically change over roundtrips (Fig. 4c, e), in sharp contrast to the case with a small average-chirp gradient (Fig. 4d, f). These results reveal the importance of the average-chirp gradient for understanding the versatile spectral-temporal dynamics over the speckled mode profile.

## Dynamics of 3D three-soliton molecules

In addition to the 3D dual-soliton molecule, we also capture 3D soliton molecules involving three solitons with equal temporal spacing, which exhibit more complicated spectral-temporal behaviors (Fig. 5, also Supplementary Fig. 22). Here, the patterns of spectral interference fringes manifest intensity variation over roundtrips (Fig. 5a, b). Such complicated spectral patterns give rise to the multiple FAC sidelobes (Fig. 5c, d), wherein the first sidelobe

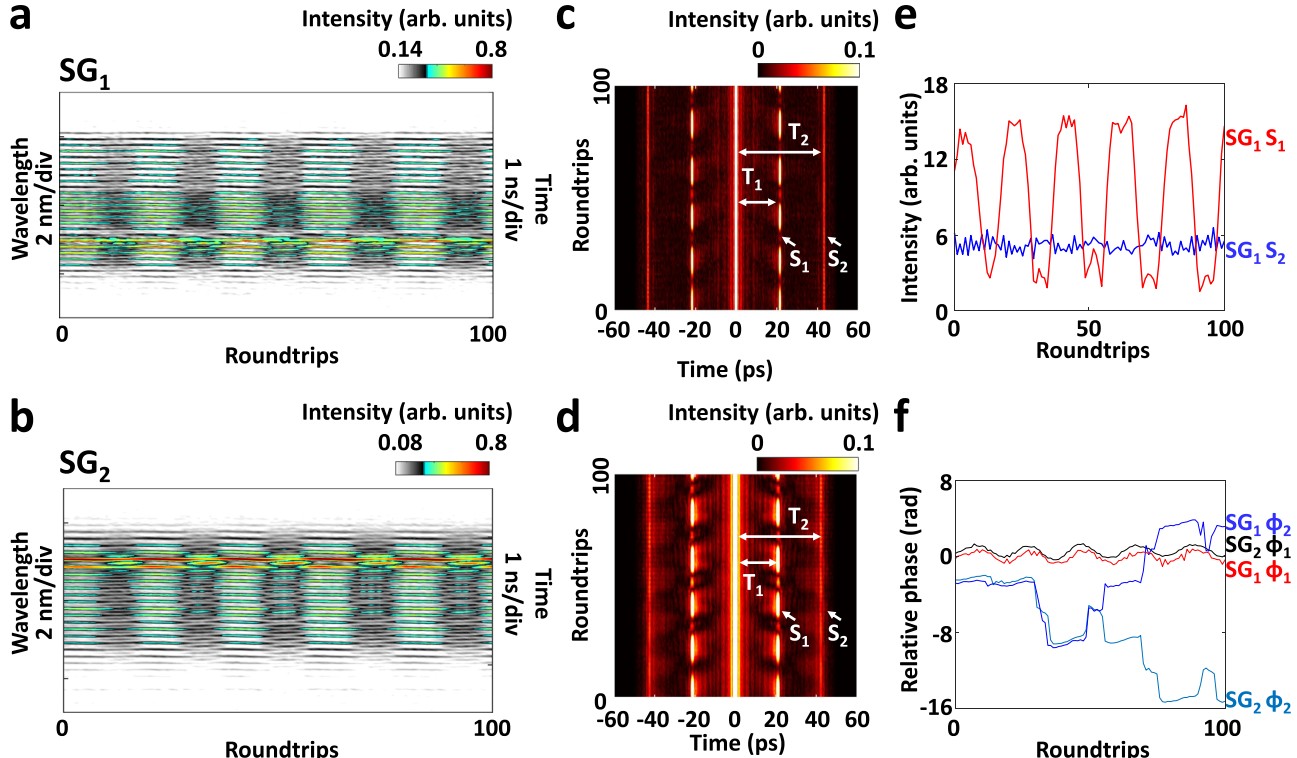

**Fig. 5 | Speckle-resolved spectral-temporal dynamics of 3D triple-soliton molecules with equal temporal spacing. a, b** Spectral evolutions in $SG_1$ and $SG_2$. **c, d** Corresponding FAC evolutions of (**a**) and (**b**). Here, $T_1$ and $T_2$ denote the temporal delays of the FAC sidelobes (i.e., $S_1$ and $S_2$). **e** Intensity variations of the FAC sidelobes $S_1$ and $S_2$ in $SG_1$. **f** Relative phase variations of the FAC sidelobes $S_1$ ($\varphi_1$) and $S_2$ ($\varphi_2$) in $SG_1$ and $SG_2$. The corresponding spectral-temporal evolutions over a longer period of time are provided in Supplementary Fig. 22.

($S_1$) exhibits obvious intensity variation, while the second sidelobe ($S_2$) maintains its intensity over roundtrips (Fig. 5e). The complicated spectral-temporal evolution can be attributed to the interaction between the solitons 1-to-2 or 2-to-3, which co-contribute to the first FAC sidelobe. The relative phases of the FAC sidelobes in different speckle grains is also calculated with acceptable accuracy (see Supplementary Note 4.2 for details), as shown in Fig. 5f. It shows that the relative phase of the FAC sidelobe $S_1$ possesses a high degree of correlation for different speckle grains, while the relative phase evolves very differently for that of the FAC sidelobe $S_2$. This can, to some extent, uncover the reason why the intensity of the first FAC sidelobe periodically changes—i.e., they maintain a consistent evolution of relative phase for various speckle grains, such that the spatiotemporal interaction can be positively accumulated.

We also observe 3D triple-soliton molecules with unequal temporal spacing under the same experimental condition, as shown in Fig. 6a–c, wherein the FAC evolution identifies the coexistence of unequal-spacing solitons. As indicated in Fig. 6a, the spectral evolution of the 3D triple-soliton molecule experiences two different regimes (i) and (ii), i.e., roundtrips 1–1100 + 1850–2000 and roundtrips 1100–1850, respectively. In regime (i), the relative phases of the FAC sidelobes vary in a small range, while the relative phases of solitons 1-to-3 ($\varphi_{13}$) and 2-to-3 ($\varphi_{23}$) sharply increase in regime (ii), as shown in Fig. 6d. The sharp variation of the relative phases $\varphi_{13}$ and $\varphi_{23}$ results in the dynamical evolution of the spectral pattern (Fig. 6b). It is also noticed that the sum of the relative phases of solitons 1-to-2 and 2-to-3, i.e., $\varphi_{12} + \varphi_{23}$, closely overlaps with that of soliton 1-to-3 ($\varphi_{13}$), i.e., the inset of Fig. 6d. The evolution trajectory in the interaction plane, i.e., Fig. 6e, reveals distinct oscillation features of the relative phase in these two regimes, wherein the oscillation amplitude of the relative phase in regime (ii) is much larger.

## Diversity of 3D soliton molecules

So far, we have presented 3D soliton molecules with versatile spatio-spectral-temporal dynamics, and yet stable 3D soliton molecules are also of great interest. Figure 7a, b illustrates the spectral-temporal evolutions of two relatively stable 3D soliton molecules, which are captured under the same experimental condition. There exists a dual-soliton molecule and an unbound soliton in Fig. 7a, while only a dual-soliton molecule in Fig. 7b. As shown in the left inset of Fig. 7a, the spectral interference fringes are twisted at a varying frequency, leading to the oscillation trajectory in a confined region of the interaction plane (Fig. 7c, e). In contrast, the trajectory extracted from Fig. 7b oscillates in three regions, as shown in Fig. 7d, f. These results highlight the importance of multi-dimensional characterization in fully quantifying the stability of the 3D laser and understanding its dynamics.

## Discussion

To summarize, we have presented the real-time spectral-temporal dynamics of 3D soliton molecules that are diverse over the speckled geometry using real-time multispeckle spectral-temporal measurement technology. The physics of the transient phenomena are experimentally and theoretically dissected, highlighting the importance of intramodal and intermodal nonlinear coupling in the dynamics of spatiotemporal mode-locking. Particularly, by correlating the real-time observations to the chemical concepts from an analogous perspective (Supplementary Fig. 23), the understandings of these interesting views can be more profound (Supplementary Note 7), e.g., the 3D dual-soliton molecules involving distinguishing transverse modes in analogy with heteronuclear diatomic molecules, and these composed of degenerate transverse modes in analogy with homonuclear diatomic molecules (Supplementary Fig. 24); furthermore, their hybrid can even give rise to more complicated evolution (Supplementary Fig. 25). We also want to stress that, on one hand, the

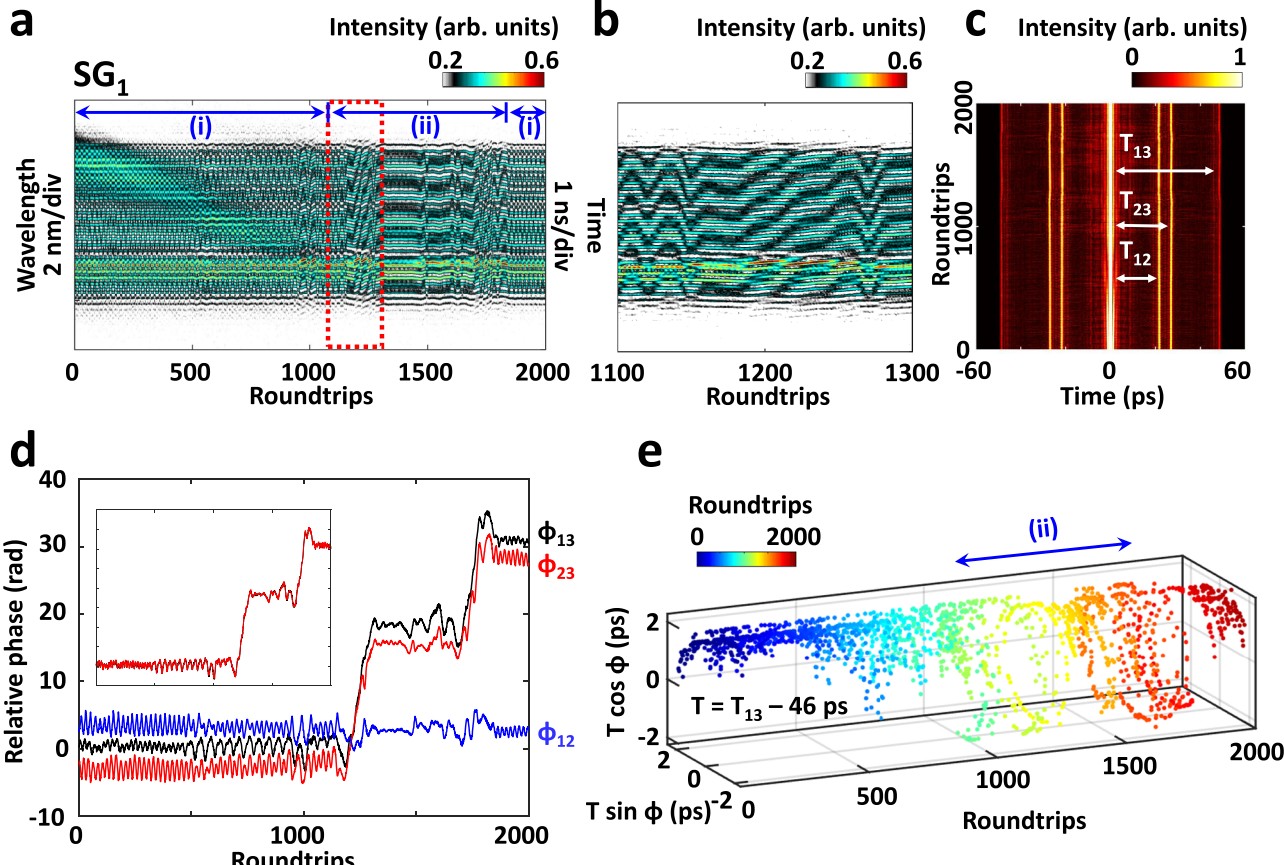

**Fig. 6 | Spectral-temporal dynamics of 3D triple-soliton molecules with unequal temporal spacing. a** Spectral evolution of the 3D triple-soliton molecule. **b** Close-up of the spectral evolution, as indicated by the red dotted square in (**a**). **c** Corresponding FAC evolution of (**a**). **d** Relative phase variations of the FAC sidelobes, i.e., solitons 1-to-2 ($\varphi_{12}$), 1-to-3 ($\varphi_{13}$), and 2-to-3 ($\varphi_{23}$), respectively. Inset shows that $\varphi_{12} + \varphi_{23}$ closely overlap with $\varphi_{13}$. **e** Evolution of the 3D triple-soliton molecule in the interaction plane. Here, only the interaction between solitons 1-to-3 is provided.

analogous interpretations of the real-time observations of 3D soliton molecules from a chemical perspective can facilitate the understanding of their behaviors and thus dissect the underlying mechanisms; on the other hand, such a platform capable of generating 3D soliton molecules and real-time multi-dimensional observation can be a powerful tool for studying complicated chemical problems, and even various multi-dimensional nonlinear problems in the fields of thermodynamics, hydrodynamics, Bose-Einstein condensates, breathers, rogue waves, etc.

## Methods
### Experimental setup
The experimental system consists of a spatiotemporal mode-locked (STML) multimode fiber laser[6,27] (Supplementary Fig. 8) and MUST measurement system (Supplementary Fig. 9), for the generation of 3D soliton molecules and multi-dimensional visualization, respectively. In brief, the STML multimode fiber laser has a ring cavity, where an Yb-doped fiber (Nufern LMA-YDF-15/130-VIII, 5 m length, 15 μm core size) serves as the gain medium. A multimode grade-index (GRIN) fiber (Thorlabs GIF 625, 2.5 m length, 62.5 μm core size) is fusion-spliced to the gain fiber with a large core offset for exciting the higher-order modes. A bandpass filter and a polarization-dependent isolator are utilized for the STML operation. The laser signal is extracted by a 50:50 beam splitter. The size of the extracted laser beam is enlarged by a ×5 magnification telescope, which is subsequently launched to the MUST measurement system. The magnification telescope associated with the single-mode fiber probe can enable a spatial resolution of -2.6 μm

(Supplementary Note 3.2). In the MUST measurement system, the laser signal is split into three branches, and three speckle grains of the multimode laser beam are individually received by the three single-mode fiber probes (while a larger number of probes is also available by compromising the compactness of the measurement system, see Supplementary Note 3.3). The collected signals are combined using optical time division multiplexing (OTDM) technology. The OTDM signal is split into two parts, one of which is directly detected by a high-speed photodiode (Newport Model 1544, 12 GHz bandwidth). The other branch is launched to a long single-mode fiber to perform real-time spectroscopy using dispersive Fourier transformation (DFT), wherein the optical signal propagates back and forth in the long single-mode fiber through a circulator and a dielectric reflector, such that a large group delay dispersion (GDD, about −0.6 ns/nm in this case) can be obtained with a moderate length of single-mode fiber. The spectroscopy signal is detected by another photodiode. The outputs of the photodiodes are finally recorded by a 4-channel real-time oscilloscope at a sampling rate of 80 GS/s.

### Data processing
The OTDM data are firstly segmented according to the roundtrip time, resulting in a two-dimensional matrix $M \times N$, in which each column ($M$) designates temporal (spectral) information, and each row ($N$) represents the roundtrip number. For the time-stretched signal (i.e., DFT), specifically, a coordinate transform is applied in terms of $\lambda = t/D_2$, where $t$ and $\lambda$ represent the retarded time and wavelength, $D_2$ is the amount of dispersion used in the DFT unit, i.e., −0.6 ns/nm. The

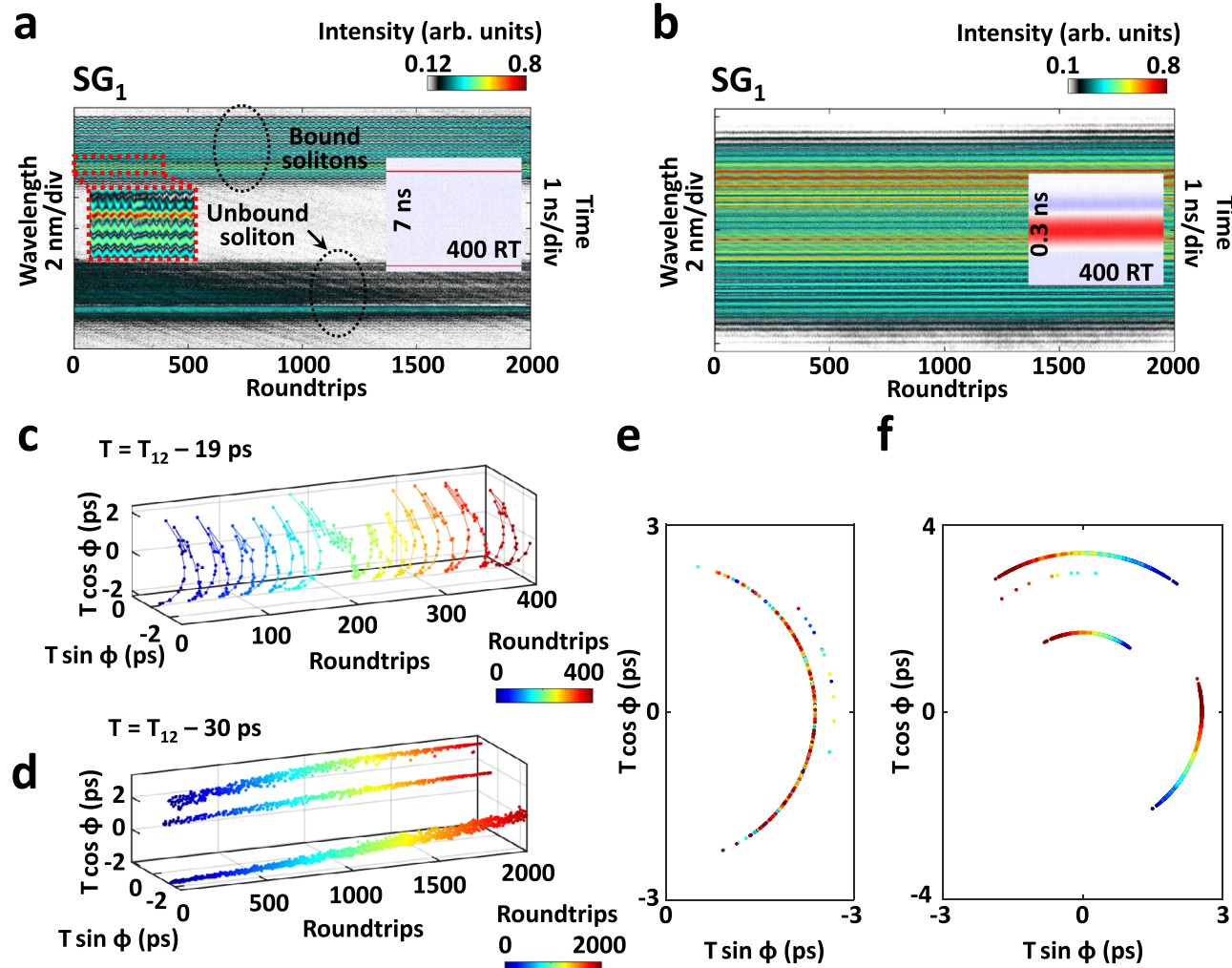

**Fig. 7 | Relatively stable 3D multi-soliton molecules with different vibration patterns. a, b** Spectral evolutions of two relatively stable 3D multi-soliton molecules in SG₁. Here, these two 3D soliton molecules exhibit different temporal and spectral signals from different speckle grains are demultiplexed by the time delays, i.e., 15.06 and 30.75 ns for SG₁-SG₂ and SG₁-SG₃, respectively.

temporal and spectral signals (right insets). Left inset of (**a**) shows the close-up of the spectral evolution. **c, d** Corresponding evolutions of the two 3D multi-soliton molecules in the interaction plane. **e, f** Projections of (**c**) and (**d**).

temporal and spectral signals from different speckle grains are demultiplexed by the time delays, i.e., 15.06 and 30.75 ns for SG₁-SG₂ and SG₁-SG₃, respectively.

**Numerical simulation**

Without loss of generality, we consider a ring laser cavity that consists of a GRIN gain fiber (50 cm length, 62.5 μm core diameter), an artificial saturable absorber, a beam splitter, a bandpass filter, and an offset space filter. It is worth noting that, in contrast to a fiber length of 5 m used in the experiment, this equivalent simulation model using a shorter fiber length can provide good enough qualitative analysis and a much faster simulation speed[30], see Supplementary Fig. 7.

The propagation of the 3D light field in the GRIN gain fiber is described by the generalized multimode nonlinear Schrödinger equations (GMMNLSEs)[31], i.e.,

$$\partial_z A_p(t;z) = i\delta\beta_0^{(p)} A_p - \delta\beta_1^{(p)} \partial_t A_p$$
$$+ \sum_{m=2}^{3} i^{m+1} \frac{\beta_m^{(p)}}{m!} \partial_t^m A_p + i\frac{n_2\omega_0}{c} \sum_{l,m,n}^{N} S_{plmn}^{K} A_l A_m A_n^*, \quad (1)$$

where $A_p(t;z)$ is the field envelope of the spatial mode $p$. $\delta\beta_0^{(p)}$ and $\delta\beta_1^{(p)}$ are the propagation constant and group velocity of the

spatial mode $p$. $\beta_m^{(p)}$ is the $m$-order dispersion coefficient. $n_2$, $\omega_0$, and $c$ are the nonlinear refractive index, center angular frequency, and speed of light, respectively. $S_{plmn}^{K}$ is the nonlinear coupling coefficient.

Then, the 3D light field gain of the GRIN gain fiber, as described in ref. 30, can be written as,

$$g(x,y,\omega;z) = \frac{g_0(\omega)}{1 + \int |A(x,y,t;z)|^2 dt/F_{sat}}, \quad (2)$$

with $A(x,y,t;z) = \sum_p \frac{F_p(x,y)A_p(t;z)}{\sqrt{\iint F_p(x,y)dxdy}}$,

where $F_{sat}$ is the saturation fluence of the multimode gain fiber, $g_0$ is the small signal gain coefficient, and $F_p(x,y)$ is the transverse-mode-field distribution of mode $p$. The saturable absorption effect is established using a transfer function after the gain fiber propagation, and then the 3D field envelope (including all the spatial modes) becomes

$$A(x,y,t;z) \rightarrow A(x,y,t;z)\sqrt{1 - (1 + |A(x,y,t;z)|^2/I_{sat})^{-1}}, \quad (3)$$

where $I_{sat}$ is the saturation intensity of the absorber.

The oscillation signal is extracted with a constant ratio, i.e.,

$$A(x,y,t;z) \rightarrow A(x,y,t;z)\sqrt{0.4} \, . \qquad (4)$$

## Gradient of the average chirp of 3D soliton molecules

For the spatial-mode-resolved and 3D light fields of 3D soliton molecules, i.e., $A_p(t;z)$ and $A(x,y,t;z)$, respectively, their average chirp $\langle\omega\rangle$ can be expressed as[32–34]:

$$\langle\omega(z)\rangle = \frac{i\int[A_p(\partial_t A_p^*) - A_p^*(\partial_t A_p)]dt}{2\int A_p^*(t;z)A_p(t;z)dt}, \qquad (5a)$$

$$\langle\omega(x,y;z)\rangle = \frac{i\int[A(\partial_t A^*) - A^*(\partial_t A)]dt}{2\int A^*(x,y,t;z)A(x,y,t;z)dt}. \qquad (5b)$$

Then, the corresponding 1D and 2D gradients of the average chirp can be defined as:

$$1D: \; \boldsymbol{\nabla}\langle\omega(z)\rangle = \frac{d\langle\omega(z)\rangle}{dz}, \qquad (6a)$$

$$2D: \; \boldsymbol{\nabla}\langle\omega(x,y;z)\rangle = \frac{\partial\langle\omega(x,y;z)\rangle}{\partial x}\vec{i} + \frac{\partial\langle\omega(x,y;z)\rangle}{\partial y}\vec{j}, \qquad (6b)$$

where $\boldsymbol{\nabla}$ is the nabla operator, $\vec{i}$ and $\vec{j}$ are the unit vectors on the $x$ and $y$ directions, respectively. More details are provided in Supplementary Note 7.

## Data availability
All data used in this study are available from the corresponding authors upon reasonable request.

## Code availability
All custom codes used in this study are available from the corresponding authors upon reasonable request.

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

## Acknowledgements

This work is partially supported by NSFC Development of National Major Scientific Research Instrument (61927816), Introduced Innovative Team Project of Guangdong Pearl River Talents Program (2021ZT09Z109), China Postdoctoral Science Foundation (2022M721197), Natural Science Foundation of Guangdong Province (2021B1515020074), Mobility Programme of the Sino-German (M-0296), Double First Class Initiative (D6211170), Guangdong Key Research and Development Program (2018B090904003), National Natural Science Foundation of China (NSFC) (U1609219), Science and Technology Project of Guangdong (2020B1212060002), and Key R&D Program of Guangzhou (Grant No. 202007020003).

## Author contributions

Y.K.G. and W.L. performed the experiments. Y.K.G., W.L., and T.L. performed numerical simulations. W.L.W. and R.S.Z. processed and analyzed the data. Y.K.G., W.L., Y.Q.X., and X.M.W. wrote the manuscript. All authors commented on the manuscript. X.M.W. and Z.M.Y. supervised the project.

## Competing interests

The authors declare no competing interests.
