## [Peer Review File · Nature Communications]

REVIEWER COMMENTS

Reviewer #1 (Remarks to the Author):

The authors reported the real-time speckle-resolved spectral-temporal dynamics of 3D soliton molecules for a long time interval using multi-speckle spectral-temporal measurement technology. They observed speckle-resolved birth, spatiotemporal interaction, and internal vibration of 3D soliton molecules. By comprehensively analyzing the observations, they found that nonlinear spatiotemporal coupling is the main origin for these dynamics, where a large average-chirp gradient over the speckled area was characterized. These findings are original, and can pave the way for further understanding the complexity of 3D soliton molecules, as well as other multidimensional nonlinear problems. The manuscript is well-organized and the results support their claims, thus I recommend it for publishing if the authors address the following concerns:

- 1) The information acquired by inspecting the gradient distribution of the average chirp is interesting. Does it indicate that the speckle grains covering the area with larger average-chirp gradient have more complex evolutionary dynamics?
- 2) As there exists intensity modulation in the FAC evolution as shown in Fig. 5, can one observe any periodic intensity variation from the corresponding temporal waveform (i.e. pulse train signal directly captured by the real time oscilloscope)?
- 3) The authors used a YDFA in the MUST system, will the performance of the YDFA affect the measurement?
- 4) In experiments, the authors used a LMA gain fiber with 5 meter length; while in simulation, the authors used a grade-index (GRIN) gain fiber with 50 cm length. Do the parameters mismatching affect the conclusions?
- 5) Lines 89-90, "... the self-stated mode-locking in three dimensions." I think it should be "the self-started mode-locking".
- 6) Line 133, "calculations for" should be "calculations from"
- 7) Line 282, Supplementary Fig. 7 illustrates the setup of the STML laser only, I suggest including Supplementary Fig. 8.
- 8) Lines 325-326, "can be rewritten" should be "can be written".
- 9) Lines 328 and 332, the authors should use 'saturation fluence of the multimode gain fiber' and 'saturation intensity of the absorber' to avoid confusion.
- 10) Please check the manuscript, all the "Insert" should be "Inset".
- 11) The authors should provide the roundtrip number of the averaged mode profile provided in Fig. 3.
- 12) Could the authors enlarge Fig. 7e and 7f for better understanding?

Reviewer #2 (Remarks to the Author):

This manuscript addressed the observations of 3D soliton molecules in a real-time way. Soliton molecules have been undergoing lots of studies, and the authors shift interests into a spatiotemporal landscape for unveiling the complexity.

For the theoretical studies, 3D soliton molecules in a multimode laser oscillator are numerically simulated and provide a promising platform for predicting the soliton dynamics. The novelty could be concluded as the observing method of real-time multispeckle spectral-temporal measurement, called MUST. See in the suppl., the MUST system includes the combination of OTDM and TS-DFT. From positive aspect, the proposed system provided an efficient approach for analyzing the dynamics of 3D molecules. Considering Ref.9, 10, it made some progress in 3D soliton molecule dynamics. However, the unveiled results still present traditional DFT assisted landscapes. I think the authors proposed an advanced and appropriate approach for continuing the research of soliton molecules in STML oscillators. More comprehensive understandings asked the authors to summarize the underlying dynamic rules beyond each observation.

By the way, this manuscript presents some interesting views for soliton dynamics. I suggest the 'unveiling' should be much more profound for inspiring audiences.

In addition, concerning some technical questions, the retrieved phase seems to be not very clear or exact. It is due to the optical detections, or the phase-retrieved method.

Reviewer #3 (Remarks to the Author):

This manuscript describes the observation of spatiotemporal soliton molecules through multi-speckle spectral-temporal measurement technology. The major advance is that the information in time, frequency, and space in solitons can be recorded at the same time. The overall method should be valid and the principle is explained clearly. However, I have several comments

1. There are other existing methods that are capable imaging complex optical processes in multiple domains. Notably the compressed ultrafast photography (Nature volume 516, pages74–77 (2014)). Soliton dynamics (Nature Communications 11,2059 (2020)) and optical chaos (Science advances 7 (3), eabc8448) have been observed. These methods should be discussed/referenced and compared with the method in the paper.

2. The spatial resolution is implemented by using a single-mode fiber as a spatial filter for the multi-mode fiber in which soliton is generated. Discussion about the spatial resolution should be included. Moreover, some information and plan about how to realize more spatial channels will be helpful for readers to understand the limit and advantage of this method.

3. The discussion to link soliton molecules with chemical and physical properties of real molecules is vague. Some references should be given in the introduction part. It may be helpful to give some concrete examples about how the nonlinear interaction in fiber optics can be mapped to molecule dynamics.

4. The authors did very careful characterization of the soliton molecule. However, it will be good if more discussion can be included about the soliton molecule dynamics. Such as what are the parameters and factors that lead to the observation of different soliton molecules. While some simulation results are given, a more intuitive explanation and clear physics picture can be helpful.

5. There are too many abbreviations in the manuscript, and some are difficult to follow. For example, SG in some other papers are normally used as Signal Generation, but here as Speckle Grain. RT can be Room Temperature, but here as Round Trip.

Overall, I believe the manuscript is novel, and can be published under revision.

**Manuscript #NCOMMS-22-40453-T: “Unveiling the complexity of
spatiotemporal soliton molecules in real time”**

Authors’ responses to reviewers’ comments

We thank the reviewers for their valuable comments and suggestions on improving our manuscript. We have carefully considered all the comments and made appropriate amendments to the manuscript. Our point-by-point responses are shown below, wherein we first list reviewers' comments in italics and then respond to them. The manuscript has also been accordingly modified, wherein the changes are highlighted in red.

Authors’ point-by-point responses to reviewers’ comments

Reviewer #1: *The authors reported the real-time speckle-resolved spectral-temporal dynamics of 3D soliton molecules for a long time interval using multi-speckle spectral-temporal measurement technology. They observed speckle-resolved birth, spatiotemporal interaction, and internal vibration of 3D soliton molecules. By comprehensively analyzing the observations, they found that nonlinear spatiotemporal coupling is the main origin for these dynamics, where a large average-chirp gradient over the speckled area was characterized. These findings are original, and can pave the way for further understanding the complexity of 3D soliton molecules, as well as other multidimensional nonlinear problems. The manuscript is well-organized and the results support their claims, thus I recommend it for publishing if the authors address the following concerns.*

Authors’ response: We thank the reviewer for the positive comments on our manuscript. Our responses to the reviewer’s concerns are as follows.

Comment 1: *The information acquired by inspecting the gradient distribution of the average chirp is interesting. Does it indicate that the speckle grains covering the area with larger average-chirp gradient have more complex evolutionary dynamics?*

Authors' response: We appreciate the reviewer for the insightful comment. The simulation results show that, over the speckled mode profile, the area with a larger average-chirp gradient has more complex evolutionary dynamics, as shown in **Fig. R1**. As can be observed, the temporal signal in the area with a larger average-chirp gradient dramatically varies over the evolution, while it exhibits small variation in the area with a smaller average-chirp gradient.

Fig. R1. Temporal evolutions along the gradient direction. **a.** 2D gradient (black arrows) of the average chirp of the 3D soliton molecule over the speckled mode profile, in roundtrip (RT) 62. **b,c.** Temporal evolutions along the white (**b**) and yellow (**c**) arrows indicated in **a**. **d,e.** Line plots of the temporal signals at the start and end points of the white (**d**) and yellow (**e**) arrows indicated in **a**.

To address the reviewer’s concern, the corresponding discussion has been provided in the revised manuscript: **The simulation results imply that, in the area with a large average-chirp gradient, e.g., the white arrow of Fig. 4a, the temporal and spectral profiles significantly vary along the gradient direction (Fig. 4c). In contrast, they remain relatively stable in the area with a small average-chirp gradient (Fig. 4d). Notably, the temporal and spectral evolutions in the area with a large average-chirp gradient dramatically change over roundtrips (Figs. 4c,e), in sharp contrast to the case with a small average-chirp gradient (Figs. 4d,f).**

Comment 2: *As there exists intensity modulation in the FAC evolution as shown in Fig. 5, can one observe any periodic intensity variation from the corresponding temporal waveform (i.e. pulse train signal directly captured by the real time oscilloscope)?*

Authors’ response: We are grateful to the reviewer for the valuable comment. As shown in **Fig. R2**, the periodic intensity variation is barely observed from the evolution of the corresponding temporal waveform directly captured by the real-time oscilloscope, i.e., **Fig. R2b**.

Fig. R2. Temporal evolution in the speckle grain SG₁. a. FAC evolution in SG₁, i.e., Fig. 5c in the main text. b. Corresponding temporal waveform captured by the real-time oscilloscope.

Comment 3: *The authors used a YDFA in the MUST system, will the performance of the YDFA affect the measurement?*

Authors' response: We thank the reviewer for his/her careful studies on our results. To avoid the potential spectral distortion from using a YDFA in the spectroscopy measurement, we have carefully optimized the parameters of the YDFA, mainly adjusting the length of the Yb-doped gain fiber used in the YDFA to suppress the nonlinear effects, which potentially occur on the forward direction of the double-pass dispersive Fourier transformation (DFT). It is also worth noting that, the YDFA imparts limited change to the spectral shape of the amplified signal on the backward direction, which is because the pulse signal has been largely broadened after double-passing the long SMF that is highly dispersive, such that its peak power is sufficiently low and the nonlinear effects are prevented.

To clarify this concern, we tested the spectroscopy performance of the MUST measurement system employing the YDFA, and the results are shown in **Fig. R3**. In this case, we used a single-mode mode-locked (SM ML) laser (**Fig. R3a**) with a repetition rate comparable to that of the STML laser utilized in the manuscript. To examine the influence of the YDFA on the spectral shape of the amplified signal — i.e., the main concern for real-time spectroscopy analysis, in each measurement we fixed the net gain of the YDFA to be ~20 dB, which is comparable to that of the loss of double-passing the long SMF (about 16 dB). Two different power levels of the input signal were tested, i.e., -8 dBm (**Fig. R3b**) and 1 dBm (**Fig. R3c**), respectively. The results show that, the optical spectrum of the amplified signal is well maintained for a low signal power level, i.e., the case of -8 dBm in **Fig. R3b**, while it could be slightly changed for a higher signal power level, i.e., 1 dBm in **Fig. R3c**. In our experiments, the power of the signal captured by the MUST measurement system is typically less than -10 dBm. As a result, measurement accuracy can be ensured, which can also be justified by other implementations of DFTs involving optical amplifiers [*Nat. Photon.* 7, 102-112 (2013)].

Fig. R3. Optical spectra collected by the standard optical spectrum analyzer (OSA) and MUST measurement system that involves a YDFA. **a.** Configuration of the dispersive Fourier transformation unit in the MUST measurement system that employs a YDFA for loss compensation. Here, the output of a single-mode mode-locked (SM ML) laser is employed as the signal under test. **b,c.** Spectroscopy measurements with different input signal powers, i.e., -8 dBm (**b**) and 1 dBm (**c**), respectively. Here, the net gain is fixed to be ~ 20 dB for both cases. The OSA curves were recorded before the YDFA, while the MUST curves were recorded after the YDFA and double-pass SMF units, i.e., right before the photodiode (PD).

To clarify this concern, we have provided the corresponding discussion in the revised Supplementary Information: **To avoid the potential spectral distortion from using a YDFA in the dispersive Fourier transformation (DFT), we have carefully optimized the parameters of the YDFA, mainly adjusting the length of the Yb-doped gain fiber used in the YDFA to suppress the nonlinear effects, which potentially occur on the forward direction. It is also worth noting that, the YDFA imparts limited change to the spectral shape of the amplified signal on the backward direction, which is because the pulse signal has been largely broadened after double-passing the long SMF that is highly dispersive, such that its peak power is sufficiently low and the nonlinear effects are prevented.**

We tested the spectroscopy performance of the MUST measurement system employing the YDFA, and the results are shown in Supplementary Figure 10. In this case, a single-mode mode-locked (SMML) laser with a repetition rate comparable to that of the STML laser utilized in the manuscript. In each measurement, we fixed the net gain of the YDFA to be ~20 dB, which is comparable to that of the loss of double-passing the long SMF (about 16 dB). Two different power levels of the input signal were tested, i.e., -8 dBm and 1 dBm, respectively. The results show that, the optical spectrum of the amplified signal is well maintained for a low signal power level, i.e., the case of -8 dBm, while it could be slightly changed for a higher signal power level, i.e., 1 dBm. In this work, the power of the speckle-resolved signal captured by the MUST measurement system is typically less than -10 dBm, such that the accuracy of the spectroscopy measurement can be ensured. (Supplementary Figure 10 has also been added in Supplementary Note 3.1)

Comment 4: *In experiments, the authors used a LMA gain fiber with 5 meter length; while in simulation, the authors used a grade-index (GRIN) gain fiber with 50 cm length. Do the parameters mismatching affect the conclusions?*

Authors' response: We thank the reviewer for his/her insightful comments. Indeed, to accelerate the computation of (3+1)-dimensional (i.e., the x , y , t and z dimensions) simulations, in the numerical simulation we used a fiber length that is shorter than that of the experimental implementation, but without loss of generality for qualitative analysis, which has also been adopted in prior works, like Ref. [*Nat. Phys.* 16, 565-570 (2020)] that studies the mechanism of the spatiotemporal mode-locking.

To address the reviewer's concern, here we present the simulation results (**Fig. R4**) by using a laser cavity the same as that of the experiment, i.e., composed of 5-m gain fiber and 4.5-m passive fiber (noted that,

there is also a small part of free space, about 30 cm). As can be observed, 3D soliton molecules can be successfully generated. In this case, the massive calculation of (3+1)-D light field $(x,y,t;z)$ takes about 3 days to finish a single simulation of generating 3D soliton molecules. Although the use of generalized multimode nonlinear Schrödinger equations (GMMNLSEs) can, to some extent, reduce the computation, it is still time-consuming for the (3+1)-D simulation of complex laser dynamics over a long evolution time, especially when many transverse modes (several tens to hundreds) are included.

To speed up the simulation, in this work all the parameters of the fiber laser cavity, including gain, nonlinearity, chromatic/modal dispersion, saturable absorption, spectral/spatial filtering, etc, have been equivalently integrated into the simulation model with a shorter fiber length, i.e., 50 cm, such that it takes only about 3 hours to finish a single simulation of generating 3D soliton molecules, which largely facilitates the numerical study of complex 3D soliton molecules. Compared with the results generated from a simulation model consistent with the experiment (**Fig. R4**), similar dynamics of 3D soliton molecules can be generated from the equivalent model used in this work, e.g., **Supplementary Figure 5**. Considering the numerical results using both consistently long fiber and equivalently short fiber (i.e., the equivalent model), the latter case can precisely predict the experiment with a much faster simulation speed, but without loss of generality, e.g., **Fig. 3** and **Supplementary Figures 3-6**.

Fig. R4. Simulated temporal evolutions of 3D soliton molecules in different speckle grains using a fiber laser cavity the same as that of the experiment. a. Temporal evolution in SG_1 . **b.** Temporal evolution in SG_2 . Here, the simulation model consists of 5-m gain fiber and 4.5-m passive fiber, which is consistent with that of the experiment.

From the above discussion, we can conclude that the equivalent simulation model using a shorter fiber length can provide good enough qualitative analysis and a much faster simulation speed. To clarify this concern, we have provided a corresponding discussion in the revised manuscript: **It is worth noting that, in contrast to a fiber length of 5 m used in the experiment, this equivalent simulation model using a shorter fiber length can provide good enough qualitative analysis and a much faster simulation speed³¹, see Supplementary Figure 7.**

And in the Supplementary Information: **As mentioned in the beginning of this section, an equivalently shorter fiber is used in the numerical simulation of this work. To identify if this equivalent simulation model using a shorter fiber length can provide good enough qualitative analysis, in contrast to the case using a consistently long fiber (i.e., the same as that of the experiment), we conduct numerical simulation by using a laser cavity the same as that of the experiment, i.e., composed of 5-m gain fiber and 4.5-m passive fiber (noted that, there is also a small part of free**

space, about 30 cm). As can be observed from Supplementary Figure 7, similar dynamics of 3D soliton molecules, like Supplementary Figure 5, are successfully generated. In this case, the massive calculation of (3+1)-D light field $(x,y,t;z)$ takes about 3 days to finish a single simulation of generating 3D soliton molecules, making a sharp contrast to that of the equivalent simulation model (taking only about 3 hours). Although the use of generalized multimode nonlinear Schrödinger equations (GMMNLSEs) can, to some extent, reduce the computation, it is still time-consuming for the (3+1)-D simulation of complex laser dynamics over a long evolution time, especially when many transverse modes (several tens to hundreds) are included. (Supplementary Figure 7 has also been added in Supplementary Note 1)

Comment 5: Lines 89-90, “... the self-stated mode-locking in three dimensions.” I think it should be “the self-started mode-locking”.

Authors’ response: We appreciate the reviewer for his/her careful reading of our manuscript. We have modified the corresponding description in the revised manuscript.

Comment 6: Line 133, “calculations for” should be “calculations from”.

Authors’ response: We have modified it in the revised manuscript.

Comment 7: Line 282, Supplementary Fig. 7 illustrates the setup of the STML laser only, I suggest including Supplementary Fig. 8.

Authors' response: We have modified this in the revised manuscript.

Comment 8: *Lines 325-326, “can be rewritten” should be “can be written”.*

Authors' response: We have modified it in the revised manuscript.

Comment 9: *Lines 328 and 332, the authors should use ‘saturation fluence of the multimode gain fiber’ and ‘saturation intensity of the absorber’ to avoid confusion.*

Authors' response: We appreciate the reviewer for his/her careful reading of our manuscript. We have modified the corresponding description in the revised manuscript.

Comment 10: *Please check the manuscript, all the “Insert” should be “Inset”.*

Authors' response: We have corrected this typo in the revised manuscript.

Comment 11: *The authors should provide the roundtrip number of the averaged mode profile provided in Fig. 3.*

Authors' response: We have correspondingly modified the figure in the revised manuscript.

Comment 12: *Could the authors enlarge Fig. 7e and 7f for better understanding?*

Authors' response: We have modified this figure in the revised manuscript (as shown in **Fig. R5**).

Fig. R5. Modified version of Fig. 7.

We hope that these revisions could satisfy the reviewer's concerns and that they meet the publication requirements. Thank you very much for your attention and consideration to our paper.

Reviewer #2: *This manuscript addressed the observations of 3D soliton molecules in a real-time way. Soliton molecules have been undergoing lots of studies, and the authors shift interests into a spatiotemporal landscape for unveiling the complexity.*

For the theoretical studies, 3D soliton molecules in a multimode laser oscillator are numerically simulated and provide a promising platform for predicting the soliton dynamics. The novelty could be concluded as the observing method of real-time multispeckle spectral-temporal measurement, called MUST. See in the suppl., the MUST system includes the combination of OTDM and TS-DFT. From positive aspect, the proposed system provided an efficient approach for analyzing the dynamics of 3D molecules. Considering Ref.9, 10, it made some progress in 3D soliton molecule dynamics. However, the unveiled results still present traditional DFT assisted landscapes. I think the authors proposed an advanced and appropriate approach for continuing the research of soliton molecules in STML oscillators. More comprehensive understandings asked the authors to summarize the underlying dynamic rules beyond each observation.

By the way, this manuscript presents some interesting views for soliton dynamics. I suggest the ‘unveiling’ should be much more profound for inspiring audiences.

In addition, concerning some technical questions, the retrieved phase seems to be not very clear or exact. It is due to the optical detections, or the phase-retrieved method.

Authors’ response: We are grateful to the reviewer for these positive and valuable comments on improving our manuscript. According to the comments from the reviewer, the deeper understandings of what we have observed are provided as follows, i.e., the major concern from the reviewer, particularly in the aspects: (1) underlying dynamic rules of these observations and profound understandings of these interesting views of 3D soliton dynamics; (2) accuracy of the phase-retrieval method.

(1) Underlying dynamic rules of these observations and profound understandings of these interesting views of 3D soliton dynamics

To provide an intuitive picture of the underlying dynamic rules of spatiotemporal (ST) soliton molecules, we first make a closely-related analogy between the photonic and chemical concepts, as outlined in **Fig. R6**, and then try to understand these observations using such a photonic-chemical analogy.

A	B		C	
LP ₀₁	^a B LP _{11a}	^b B LP _{11b}	^a C LP _{21a}	^b C LP _{21b}
				D	E		F	
LP ₀₂	^a E LP _{31a}	^b E LP _{31b}	^a F LP _{12a}	^b F LP _{12b}
				
Notation	Chemical analogy	Photonic concept
X (A-F)	Element	LP modes
^a X, ^b X	Isotopes	Degenerate modes
X ₂	Homonuclear diatomic molecule	Soliton molecule with a (degenerate) mode(s)
XY	Heteronuclear diatomic molecule	Soliton molecule with distinguishing modes
X—Y	Bond strength	Average-chirp gradient

Fig. R6. Multimode elements and concept of the photonic-chemical analogy.

3D dual-soliton molecule

In comparison with 1D soliton molecules with pulse-to-pulse interaction resulted from modal-irrelevant effects (e.g., $\chi^{(3)}$ nonlinear effect [*Phys. Rev. A* 78, 063817 (2008)], dynamic gain [*IEEE J. Quant. Electron.* 34, 1749 (1998)], dispersive-wave emission [*Nat. Commun.* 10, 5756 (2019)], acoustic-wave effect [*Nat. Photon.* 10, 454 (2016)]), 3D dual-soliton molecules exhibit a much greater complexity. Particularly, their assembling dynamics are governed by the spatiotemporal interaction involved

intermodal cross-phase modulation (IM-XPM), intermodal four-wave mixing (IM-FWM), and modal dispersion. To understand these observed dynamics from the perspective of the photonic-chemical analogy shown in **Fig. R6**, here we discuss two typical kinds of 3D dual-soliton molecules (**Fig. R7**): **the first kind is made up of two different linearly-polarized (LP) modes (denoted as kind I); the second kind is constituted by a (degenerate) mode (denoted as kind II).**

For kind-I dual-soliton molecules, the pulses in two distinguishing transverse modes initially propagate at different group velocities due to the modal dispersion of the multimode fiber. Once they overlap in the time domain, it gives rise to the spatiotemporal interaction through the interplay of IM-XPM and IM-FWM [*Nat. Commun.* 4, 1719 (2013); *Nat. Photon.* 9, 306 (2015); *Opt. Express* 23, 3492 (2015)], as shown in **Fig. R7a**. This process can be described by the generalized nonlinear Schrödinger equation [*Opt. Lett.* 42, 3419 (2017)], i.e.,

$$\frac{\partial A_p}{\partial z} = -i \frac{\beta_{21}}{2} \frac{\partial^2 A_p}{\partial t^2} + i \gamma_{pp} |A_p|^2 A_p + \underbrace{2i \gamma_{pq} |A_p|^2 A_q}_{IM-XPM} + \underbrace{i \gamma_{pq} A_p^2 A_q^* \exp(-2i\Delta\beta z)}_{IM-FWM},$$

$$\text{with } \gamma_{pp} = \frac{n_2 \langle \omega_1 \rangle Area_p}{c}, \gamma_{pq} = \frac{n_2 \langle \omega_1 \rangle}{c} \frac{\iint_{-\infty}^{\infty} |F_p(x, y)|^2 |F_q(x, y)|^2 dx dy}{\iint_{-\infty}^{\infty} |F_p(x, y)|^2 dx dy \iint_{-\infty}^{\infty} |F_q(x, y)|^2 dx dy},$$

where $A_p(t; z)$ and $A_q(t; z)$ are the field envelopes of the LP_{21a} and LP_{31a} modes, respectively; while $F_p(x, y)$ and $F_q(x, y)$ are the corresponding transverse-mode-field distributions. $Area_p$ and β_{21} are the effective mode area and second-order dispersion of the LP_{21a} mode, respectively. n_2 is the nonlinear refractive index. $\Delta\beta$ represents the propagation constant mismatch, and c is the speed of light. To intuitively understand the mechanism of providing the binding force, we try to interpret it through an analogous ‘chemical bond’ generated between two distinguishing atoms (e.g., ^aC for LP_{21a} and ^aE for LP_{31a}

as indicated in **Fig. R7a**) by means of the Gordon-Mollenauer approach [*Phys. Rev. A* 78, 063817 (2008)], and the force can be expressed as the 1D gradient of the average chirp $\langle\omega_1\rangle$ (also see **Methods**), i.e.,

$$f = \frac{d\langle\omega_1\rangle}{dz} = \frac{i}{2W} \int_{-\infty}^{+\infty} dt \left[\frac{d}{dz} (A_p \partial_t A_p^* - A_p^* \partial_t A_p) \right], \quad W = \int_{-\infty}^{+\infty} |A_p|^2 dt.$$

By combining it with the equation of $\partial A_p / \partial z$ mentioned before, we have

$$f = \frac{\gamma_{pq}}{W} \int_{-\infty}^{+\infty} dt \{ \text{Re}(A_p \partial_t A_p^*) \text{Re}[A_p^* A_q(t - \delta_{pq}z; z) \times (2 + e^{2i\Delta\beta z})] \\ - \text{Im}(A_p \partial_t A_p^*) \text{Im}[A_p^* A_q(t - \delta_{pq}z; z) \times (3e^{2i\Delta\beta z} - 2)] \},$$

$$\text{with } \delta_{pq} = \beta_{11}(\langle\omega_1\rangle) - \beta_{12}(\langle\omega_2\rangle) + \beta_{21} \times \langle\omega_1\rangle - \beta_{22} \times \langle\omega_2\rangle,$$

where δ_{pq} accounts for the group velocity mismatch between LP_{21a} and LP_{31a} modes. β_{11} and β_{12} are the first-order dispersions of LP_{21a} and LP_{31a} modes, respectively. β_{22} and $\langle\omega_2\rangle$ are the second-order dispersion and average chirp of LP_{31a} mode.

Fig. R7. Temporal dynamics of two different 3D dual-soliton molecules. **a,b.** Temporal evolution (**a**), corresponding variations of average chirp $\langle \omega_1 \rangle$ (top panel of **b**) and binding force (bottom panel of **b**) for the LP_{21a} mode in the case that a dual-soliton molecule in analogy with heteronuclear diatomic molecule is produced. Here, a negative force represents an attraction force, while a repulsion force for a positive value. Insets of **a** illustrate the isosurface plots of the 3D dual-soliton molecule before and after the assembling. **c,d.** Temporal evolution and 3D view of another dual-soliton molecule in analogy with homonuclear diatomic molecule.

The calculated results are shown in **Fig. R7b**, wherein the binding force reaches its maximum during the reaction (as marked by the arrow in **Fig. R7b**), resulting in the assembling of molecule ‘CE’. It is noteworthy that this 3D dual-soliton molecule, as a photonic counterpart of heteronuclear diatomic molecule, is only accessible in the ST mode-locking that involves diverse multimode elements.

For kind-II dual-soliton molecules, the pulses in a pair of degenerate modes (e.g., LP_{31a} and LP_{31b} as indicated in **Fig. R7c**) have very similar characteristics, particularly their propagation constant, and they evolve in a landscape akin to these 1D soliton molecules, while their synchronization can be further reinforced by the maximum gain principle in the ST mode-locking [*Nat. Phys.* 16, 565 (2020); *Nat. Commun.* 12, 67 (2021)]. In contrast to kind I, kind-II 3D soliton molecules resemble homonuclear diatomic molecules. Despite its similarity with these 1D counterparts, the binding mechanism driven by the modal degeneracy and the maximum gain principle still has no analogue in the scenario of single-mode mode-locking.

Fig. R8. Temporal dynamics of 3D triple-soliton molecules. **a.** Temporal evolution of the 3D triple-soliton molecule. **b.** Energy distributions of the multimode elements before (left) and after (right) the formation of the 3D triple-soliton molecule. **c.** Visualization of the 3D triple-soliton molecule. (Top) Isosurface plot of the 3D triple-soliton molecule. (Bottom) Mode-resolved intensity profiles on the time-intensity plane.

3D triple-soliton molecule

The generation of 3D triple-soliton molecules in the ST mode-locking is much more complicated due to the modal/spatial complexity [*Nat. Commun.* 10, 1638 (2019)]. To show this, we present typical assembling dynamics in **Fig. R8a**. As illustrated, in the initial state, a single 3D soliton that involves LP_{21a}, LP_{12a} and LP_{12b} modes approaches a kind-II dual-soliton molecule (i.e., a homonuclear diatomic molecule involving LP_{31a} and LP_{31b}). As depicted in **Figs. R8a,b**, the production of the 3D triple-soliton molecule, as a hybrid case of kind-I and -II dual-soliton molecules, undergoes a similar binding process as that of the dual-soliton molecule (**Fig. R7**). Such assembling dynamics can emulate a combination reaction of generating a compound, as the invariant modal energy distribution can serve as the analogue of the law of conservation of mass. According to the mode-resolved intensity profiles shown in **Fig. R8c**, the 3D triple-soliton molecule can be treated as an analogous molecular formula CF₂E₂.

(2) Accuracy of the phase-retrieval method

To address the reviewer's concern on the accuracy of the phase-retrieval method, we first present the numerical validation of the phase-retrieval method and then discuss the phase ambiguity of the optical detection.

Numerical validation of the phase-retrieval method

To address the reviewer's concern on the validation of the phase-retrieval method used in this work, we numerically create interferograms of soliton molecules with preset relative phases ϕ , as shown in **Fig. R9a**, and subsequently evaluate the accuracy of the retrieved relative phases using these preset values, as shown in **Fig. R9b**. As can be observed, a good agreement is obtained, theoretically confirming the feasibility of the phase-retrieval method.

Fig. R9. Numerical validation of the phase-retrieval method. **a.** Interferogram of soliton molecules with preset random relative phases. **b.** Retrieved relative phases (black circles) and the preset values (red dots).

Phase ambiguity of the optical detection

In the experimental implementation, the limited bandwidth of the detection system (mainly including the photodetector and digitizer) can impose phase error $\delta\phi$ on the retrieved phase [*Phys. Rev. Lett.* **121**, 023905 (2018)], which is given as

$$\delta\phi \sim \delta\omega_{res}\tau$$

with temporal separation τ and spectral resolution $\delta\omega_{res}$ that is written as [*Opt. Express* **18**, 10016 (2010)]

$$\delta\omega_{res} = \frac{0.35}{D_2 f_{det}},$$

where, f_{det} is the bandwidth of the detection system, i.e., 21 GHz in this case (defined by the digitizer).

The phase ambiguity in this work is comparable to prior works (**Table R1**).

To further identify the feasibility of the phase evaluation, we calculate and simulate the roundtrip-evolved retrieved phase for **Fig. 6**. As shown in **Figs. R10a,b**, the numerical simulation can well reproduce varying features of ϕ_{12} , ϕ_{23} , and ϕ_{13} , e.g., the jump and plateau over the phase evolution (as indicated by the arrows). As marked by the black dashed square in **Fig. R10a**, a non-trivial fluctuation of ϕ_{12} is observed, also see the close-up shown in **Fig. R10c**.

Table R1. Phase ambiguity of real-time spectral interferometry

Temporal separation τ (ps)	D_2 (ps ²)	Bandwidth f_{det} (GHz)	Phase error $\delta\phi/2\pi$	Ref.
<0.6	18.8	~8	<22.2%	Science , 356, 50 (2017)
~2 ps	163	~6	~11.4%	PRL , 118, 243901 (2017)
40 ps	178	45	27.8%	OE , 29, 16362 (2021)
10 ps	235	20	11.9%	OE , 30, 21931 (2022)
~30 ps	360	21	~22%	This work

Fig. R10. Evolutions of the retrieved phase. **a,b.** Evolutions of the relative phase retrieved from the experimental (**a**) and simulated (**b**) interferograms. Arrows indicate the similar features shown in the experiment and simulation. **c.** Close-up of ϕ_{12} , indicated by the black dashed square in **a**.

From the above discussion, we can conclude that: **(i)** the phase-retrieval method is numerically validated; **(ii)** the bandwidth limitation of the detection system can impart an acceptable phase ambiguity for qualitative study, wherein the major feature of the phase evolution can be retrieved.

To address the reviewer's concern, we have provided a corresponding discussion in the main text of the revised manuscript: **Particularly, by correlating the real-time observations to the chemical concepts from an analogous perspective (Supplementary Figure 23), the understandings of these interesting views can be more profound (Supplementary Note 7), e.g., the 3D dual-soliton molecules involving distinguishing transverse modes in analogy with heteronuclear diatomic molecules, and these composed of degenerate transverse modes in analogy with homonuclear diatomic molecules**

(Supplementary Figure 24); furthermore, their hybrid can even give rise to more complicated evolution (Supplementary Figure 25). We also want to stress that, on one hand, the analogous interpretations of the real-time observations of 3D soliton molecules from a chemical perspective can facilitate the understanding of their behaviors and thus dissect the underlying mechanisms; on the other hand, such a platform capable of generating 3D soliton molecules and real-time multi-dimensional observation can be a powerful tool for studying complicated chemical problems, and even various multi-dimensional nonlinear problems in the fields of thermodynamics, hydrodynamics, Bose-Einstein condensates, breathers, rogue waves, etc.

Two new sections have also been added to the Supplementary Information, i.e., **Supplementary Note 7: Underlying dynamic rules of these observations and profound understandings of these interesting views of 3D soliton dynamics** and **Supplementary Note 4.2: Accuracy of the phase-retrieval method**.

More discussions have also been provided in the responses to **Comments 3,4** of **Reviewer #3**.

We hope that these revisions could satisfy the reviewer's concerns and that they meet the publication requirements. Thank you very much for your attention and consideration to our paper.

Reviewer #3: *This manuscript describes the observation of spatiotemporal soliton molecules through multi-speckle spectral-temporal measurement technology. The major advance is that the information in time, frequency, and space in solitons can be recorded at the same time. The overall method should be valid and the principle is explained clearly. However, I have several comments.*

Authors' response: We thank the reviewer for the efforts on improving our manuscript. Our point-by-point responses to the reviewer's comments are presented as follows.

Comment 1: *There are other existing methods that are capable imaging complex optical processes in multiple domains. Notably the compressed ultrafast photography (Nature volume 516, pages74–77 (2014)). Soliton dynamics (Nature Communications 11,2059 (2020)) and optical chaos (Science advances 7 (3), eabc8448) have been observed. These methods should be discussed/referenced and compared with the method in the paper.*

Authors' response: We appreciate the reviewer for his/her valuable comments. 3D solitons and their molecule-fashion can exhibit complicated spatio-spectral-temporal dynamics, due to spatiotemporal coupling and nonlinear interaction. The evolution of these dynamics can last for 1000s of roundtrips/frames, corresponding to a time scale of 10's-to-100's μ s, as shown in **Figs. 6,7** as well as **Supplementary Figures 1,3,18,22**. From this perspective, multi-dimensional measurement technologies with both fast refresh rate and long recording length/time are highly desired for studying these dynamics of 3D soliton molecules. As mentioned by the reviewer, the compressed ultrafast photography (CUP) technology [*Nature 516, 74-77 (2014)*] is a powerful tool for spatiotemporal characterization of transient events with a femtosecond-to-picosecond temporal resolution, and significant findings have been obtained

in the field of soliton dynamics [*Nat. Commun.* 11, 2059 (2020)] and optical chaos [*Sci. Adv.* 7, eabc8448 (2021)]. In short, the CUP technology works by encoding the transient scenes with pseudo-random pattern, which is followed by a shearing operation in the time domain using a streak camera with a fully opened aperture. The CUP technology can capture non-repetitive time-evolving events with a refresh rate of 10^{11} -to- 10^{13} frames per second, which is prominent for measuring x - y - t (x, y , spatial coordinates; t , time) scenes with super-fine temporal resolution. It is also noticed that, the number of continuous recording frames of the CUP technology is from 10s to 100s, and it may prevent the recording of the long-term evolving optical events, like the formation of 3D soliton molecules in this case. To this end, the MUST technology used in this work compromises the temporal resolution to a moderate level and thus prolongs the recording length (**Table R2**), especially useful for studying the long-term evolving 3D soliton molecules.

Table R2 Key parameters of the CUP and MUST technologies

Technology	If single shot	If burst mode	Temporal resolution	No. of continuous frames
CUP	Yes	Yes	100s fs to ps	10s to 100s
MUST	Yes	No	12.5 ps and 47.8 ns*	>260,000**

*Here, the temporal resolution in the time domain is defined by the sampling rate of the real-time oscilloscope (80 GS/s), while the temporal resolution in the frequency domain is defined by the repetition rate of the laser (20.9 MHz), i.e., 12.5 ps and 47.8 ns, respectively.

**The number of continuous frames is defined by the repetition rate of the laser (20.9 MHz) and memory depth of the real-time oscilloscope (1 Gpts/channel).

To address the reviewer's concern, we have cited and discussed the CUP technology in the revised manuscript: **In the meantime, super-high-speed photography technologies, especially the compressed**

ultrafast photography (CUP)²⁴, have been successfully demonstrated, and interesting findings have been obtained in the field of soliton dynamics²⁵ and optical chaos²⁶.

Comment 2: *The spatial resolution is implemented by using a single-mode fiber as a spatial filter for the multi-mode fiber in which soliton is generated. Discussion about the spatial resolution should be included. Moreover, some information and plan about how to realize more spatial channels will be helpful for readers to understand the limit and advantage of this method.*

Authors' response: We thank the reviewer for the insightful comments. For collecting the spectral-temporal signal from a single speckle grain, a single-mode fiber that can serve as spatial filter is useful. To do so, we utilize three probes to collect the spectral-temporal signals of three different speckle grains, and each probe consists of a single-mode fiber and a collimator for light coupling (**Fig. R11**). To freely access different speckle grains, the collimator is mounted on the translation stage. It is worth noting that, each bright speckle grain is a coherent spot resulted from the constructive interference. **In this sense, a spatial resolution that can resolve the speckle grains is good enough for the MUST measurement, as shown in Fig. R12.** To ensure this capability, the collimator is associated with a telescope for magnifying the size of the speckle grains, and such a configuration can provide a spatial resolution of about 2.6 μm .

Fig. R11. Configuration of collecting lights from different speckle grains using single-mode fibers. Here, the collimator is mounted on a translation stage.

Fig. R12. Speckle grains of the STML multimode fiber laser. **a.** Beam profile captured by a CCD camera. **b.** Beam profile shined on a photosensitive card.

To demonstrate the ability of simultaneously measuring more speckle grains, we implement a MUST measurement system with six channels, and the spectral evolutions of six different speckle grains are simultaneously captured, as shown in **Fig. R13**. This implies that, the channel scaling is feasible by using free-space optics, if compactness compromise is not a concern. It should also be pointed out that, here the real-time DFT spectroscopy is performed in a single long-SMF unit that is associated with the OTDM

technology. Thus, temporal overlapping of the DFT signal will limit the number of channels, i.e., about 8 channels for the configuration used in this work.

Fig. R13. MUST measurement system with six channels. a. Schematic diagram. BS: beam splitter. C: circulator. Col: collimator. L: lens. M: mirror. OC: optical coupler. ODL: optical delay line. SG: speckle grain. b-g. Spectral evolutions of six different speckle grains captured by the six-channel MUST measurement system.

To solve this problem, parallel detection with more DFT units and digitizers is helpful. In addition, using micro-lens array and multicore dispersive fiber, as shown in **Fig. R14**, can potentially further increase the

number of channels, but making the measurement system more complicated and expensive. Based on the above discussion, we implemented a MUST measurement system with only three channels in this study, given that resolving the spectral-temporal signal for three speckle grains is already useful for probing the spatial-spectral-temporal dynamics of the 3D soliton molecule, from the perspective of qualitative analysis.

Fig. R14. Potential scheme for speckle-resolved optical collection using micro-lens array and multicore dispersive fiber.

L: lens. OTDM: optical time division multiplexing.

To address the reviewer's concern, we have correspondingly discussed in the revised manuscript: **The laser signal is extracted by a 50:50 beam splitter. The size of the extracted laser beam is enlarged by a $5\times$ magnification telescope, which is subsequently launched to the MUST measurement system. The magnification telescope associated with the single-mode fiber probe can enable a spatial resolution of $\sim 2.6\ \mu\text{m}$ (Supplementary Note 3.2). In the MUST measurement system, the laser signal is split into three branches, and three speckle grains of the multimode laser beam are individually received by the three single-mode fiber probes (while a larger number of probes is also available by compromising the compactness of the measurement system, see Supplementary Note 3.3).**

Two new sections have also been added to the Supplementary Information, i.e., **Supplementary Note 3.2: Spatial resolution of the MUST measurement system** and **Supplementary Note 3.3: Channel scaling of the MUST measurement system.**

Comment 3: *The discussion to link soliton molecules with chemical and physical properties of real molecules is vague. Some references should be given in the introduction part. It may be helpful to give some concrete examples about how the nonlinear interaction in fiber optics can be mapped to molecule dynamics.*

Authors' response: We thank the reviewer for the insightful comment. To clarify the reviewer's concern, we first discuss the influence of nonlinear interaction on 3D soliton molecular dynamics over multiple dimensions. Then, the photonic-chemical analogy is established for correlating the 3D soliton molecule to real molecules, based on which molecular-like assembling dynamics are demonstrated.

Influence of nonlinear interaction on 3D soliton molecular dynamics

The dynamics of soliton molecules have been found to share many common features with the real molecular dynamics, like molecular vibration [*Science* 356, 50 (2017); *Phys. Rev. Lett.* 118, 243901 (2017)], excitation [*Nat. Photonics* 14, 9 (2020)], and on-demand synthesis [*Light Sci. Appl.* 10, 120 (2021); *Optica* 9, 240-250 (2022)]. In the one-dimensional (1D) scenario, the XPM- and FWM-mediated nonlinear interactions between pulses can create an effective trapping potential [*Phys. Rev. Lett.* 123, 243905 (2019)], and can act as a molecular-like binding force to produce soliton molecules in fiber optics [*Phys. Rev. A* 78, 063817 (2008)]. However, the existing photonic-chemical analogy is mainly applicable to 1D solitons (i.e., these existed in the fundamental transverse mode LP₀₁ using single-mode fibers), leaving the connection between higher-dimensional nonlinear interaction and real molecular dynamics largely unexplored.

To extend the existing theoretical framework for studying 3D soliton molecules, we consider a prime pulse with field envelope $A_p(t; z)$ (corresponding to transverse mode p) that is perturbed by another pulse $A_q(t; z)$ in a distinguishing transverse mode q . In this case, the nonlinear interaction between pulses in

two distinguishing transverse modes can be described by a generalized nonlinear Schrödinger equation in the form of

$$\frac{\partial A_p}{\partial z} = -i \frac{\beta_{21}}{2} \frac{\partial^2 A_p}{\partial t^2} + i \gamma_{pp} |A_p|^2 A_p + \underbrace{2i \gamma_{pq} |A_p|^2 A_q}_{\text{IM-XPM}} + \underbrace{i \gamma_{pq} A_p^2 A_q^* \exp(-2i\Delta\beta z)}_{\text{IM-FWM}},$$

$$\text{with } \gamma_{pp} = \frac{n_2 \langle \omega_1 \rangle \text{Area}_p}{c}, \gamma_{pq} = \frac{n_2 \langle \omega_1 \rangle}{c} \frac{\iint_{-\infty}^{\infty} |F_p(x, y)|^2 |F_q(x, y)|^2 dx dy}{\iint_{-\infty}^{\infty} |F_p(x, y)|^2 dx dy \iint_{-\infty}^{\infty} |F_q(x, y)|^2 dx dy},$$

where Area_p and β_{21} are the effective mode area and second-order dispersion of the prime transverse mode p , respectively. n_2 is the nonlinear refractive index. $F_p(x, y)$ and $F_q(x, y)$ are the transverse-mode-field distributions of p and q modes, respectively. $\Delta\beta$ represents the propagation constant mismatch, and c is the speed of light. Please note that, the main perturbations (in red) on the field envelope $A_p(t; z)$ are ascribed by the intermodal cross-phase modulation (IM-XPM) and intermodal four-wave mixing (IM-FWM) effects.

Through the Gordon-Mollenauer approach [*Opt. Lett.* 17, 1575-1577 (1992); *Phys. Rev. A* 78, 063817 (2008)], molecular-like binding force f can be defined by the 1D gradient of the average chirp (also see **Methods**), i.e.,

$$f = \frac{d\langle \omega_1 \rangle}{dz} = \frac{i}{2W} \int_{-\infty}^{+\infty} dt \left[\frac{d}{dz} (A_p \partial_t A_p^* - A_p^* \partial_t A_p) \right], \quad W = \int_{-\infty}^{+\infty} |A_p|^2 dt.$$

The binding force f accounting for the IM-XPM and IM-FWM effects is

$$f = \frac{\gamma_{pq}}{W} \int_{-\infty}^{+\infty} dt \{ \text{Re}(A_p \partial_t A_p^*) \text{Re}[A_p^* A_q(t - \delta_{pq} z; z) \times (2 + e^{2i\Delta\beta z})] \\ - \text{Im}(A_p \partial_t A_p^*) \text{Im}[A_p^* A_q(t - \delta_{pq} z; z) \times (3e^{2i\Delta\beta z} - 2)] \},$$

$$\text{with } \delta_{pq} = \beta_{11}(\langle \omega_1 \rangle) - \beta_{12}(\langle \omega_2 \rangle) + \beta_{21} \times \langle \omega_1 \rangle - \beta_{22} \times \langle \omega_2 \rangle,$$

where δ_{pq} accounts for the group velocity mismatch between transverse modes p and q . β_{11} and β_{12} are first-order dispersions of transverse modes p and q , respectively. β_{22} and $\langle\omega_2\rangle$ are the second-order dispersion and average chirp of mode q .

Consequently, the connection between spatiotemporal interaction in multimode fiber and molecular dynamics can be established.

A	B		C	
LP ₀₁	^a B	^b B	^a C	^b C
	LP _{11a}	LP _{11b}	LP _{21a}	LP _{21b}
				D	E		F	
LP ₀₂	^a E	^b E	^a F	^b F
	LP _{31a}	LP _{31b}	LP _{12a}	LP _{12b}
				
Notation	Chemical analogy	Photonic concept
X (A-F)	Element	LP modes
^a X, ^b X	Isotopes	Degenerate modes
X ₂	Homonuclear diatomic molecule	Soliton molecule with a (degenerate) mode(s)
XY	Heteronuclear diatomic molecule	Soliton molecule with distinguishing modes
X—Y	Bond strength	Average-chirp gradient

Fig. R15. Multimode elements and concepts for the photonic-chemical analogy.

Photonic-chemical analogy

Based on the above discussion, we can make a closely-related analogy between the photonic and chemical concepts, as shown in **Fig. R15**, wherein multimode elements (A-F) and photonic isotopes are defined. Using such a photonic-chemical analogy, we discuss a paradigm — assembling dynamics of a heteronuclear diatomic molecule ‘CE’ (i.e., a 3D dual-soliton molecule made up of LP_{21a} and LP_{31a} modes), as shown in **Fig. R16a**.

Fig. R16. 3D soliton molecular dynamics analogous to the assembling of heteronuclear diatomic molecule. **a.** Temporal evolutions of elements ${}^a\text{C}$ (LP_{21a}, top) and ${}^a\text{E}$ (LP_{31a}, bottom). **b.** Three phases of assembling dynamics of ‘CE’ molecule. The spatiotemporal interaction between elements C and E is intuitively illustrated by isosurface plots of the multimode intensity profile. **c.** Binding force. **d.** Intensity profiles on the time-intensity plane before and after the assembling (corresponding to roundtrips 1 and 1060).

Three phases of the assembling dynamics are illustrated in **Fig. R16b**. In phase 1 (P1, i.e., dissociated atoms), the pulses in LP_{21a} and LP_{31a} modes propagate at different group velocities, and there is no spatiotemporal overlapping, leading to weak nonlinear-interaction-induced binding force. Thus, the pulses in LP_{21a} and LP_{31a} modes behave like dissociated atoms, i.e., elements ${}^a\text{C}$ and ${}^a\text{E}$. The existence of group velocity mismatch, primarily caused by the modal and chromatic dispersion, can impose weak attraction on the atoms and thus initiate the self-organization of the ‘CE’ molecule.

In phase 2 (P2, i.e., binding process), the desynchronized pulses overlap in the time and space domains and thus nonlinearly interact to produce molecular-like binding force between the initially uncorrelated

atoms (i.e., dissociated elements ${}^a\text{C}$ and ${}^a\text{E}$). As shown in **Fig. R16c**, the binding process (like reaction) starts from roundtrip ~ 700 and lasts for ~ 360 roundtrips, during which a maximum strength of $f = 4.3 \times 10^{-4}$ is generated.

In phase 3 (P3, i.e., molecule formation), the equilibrium of the reaction attains is established, i.e., $d\langle\omega_1\rangle/dN = 0$, leading to the successful assembling of the heteronuclear diatomic molecule ‘CE’, whose intensity profiles on the time-intensity plane before and after the assembling are shown in **Fig. R16d**.

To address the reviewer’s concern, we have correspondingly discussed in the revised manuscript: **While experimentally observing the real-time motion of atoms and molecules is still difficult³⁻⁵, it can be studied by referencing the dynamics of three-dimensional (3D) soliton molecules in nonlinear optical systems, which may share many common characteristics with the dynamics of chemical molecules⁶; while one-dimensional (1D) soliton generated through nonlinear interaction in fiber optics has been intensively studied and correlated to chemical molecular dynamics⁷⁻⁹.**

A new section has also been added to the Supplementary Information, i.e., **Supplementary Note 7: Underlying dynamic rules of these observations and profound understandings of these interesting views of 3D soliton dynamics.**

More discussions have also been provided in the responses to **Comments of Reviewer #2**.

Comment 4: *The authors did very careful characterization of the soliton molecule. However, it will be good if more discussion can be included about the soliton molecule dynamics. Such as what are the parameters and factors that lead to the observation of different soliton molecules. While some simulation results are given, a more intuitive explanation and clear physics picture can be helpful.*

Authors' response: We thank the reviewer for the valuable suggestions. In addition to what has been discussed in the response to **Comment 3**, we further discuss 3D soliton molecular dynamics with greater complexity and subsequently the parameters affecting the state of 3D soliton molecules.

3D soliton molecular dynamics with greater complexity

In our simulation, because of the modal/spatial complexity, we observe different 3D soliton molecules in a long-term assembling process ($\sim 30,000$ roundtrips), as illustrated in **Figs. R17a,b**. Such an evolution can emulate a combination reaction of generating a compound, as the invariant modal energy distribution (**Fig. R17c**) can serve as the analogue of the law of conservation of mass. During the assembling process, different soliton molecules are discovered (**Fig. R17b**), which can be analogous to multiple intermediates produced in a complex combination reaction. In the initial state, a pair of pulses in LP_{31a} and LP_{31b} modes co-propagate as a soliton molecule. Here, we classify the dual-soliton molecule comprised of degenerate modes (i.e., photonic isotopes) as the kind-II dual-soliton molecule — an analogue of homonuclear diatomic molecule (see the responses to the comments of Reviewer #2). Individual pulses in LP_{31a} and LP_{31b} modes are synchronized because of similar propagation constant, and also it can be further reinforced by the maximum gain extraction in the ST mode-locking [*Nat. Phys.* 16, 565 (2020); *Nat. Commun.* 12, 67 (2021)]. According to the photonic-chemical analogy described in **Fig. R15**, this dual-soliton molecule can be treated as an analogous molecular formula, i.e., E₂. Then, a single 3D soliton that involves LP_{21a}, LP_{12a} and LP_{12b} modes (indicated by the blue arrow in **Fig. R17a**) approaches the homonuclear diatomic molecule E₂, resulting in the production of a 3D triple-soliton molecule, as a hybrid case of kind-I and -II dual-soliton molecules. This 3D triple-soliton molecule can be treated as an analogous molecular formula, i.e., CF₂E₂. Finally, by further reacting with a single soliton in LP_{21b} mode, as indicated by the red arrow in **Fig. R17a**, a stable molecule C₂F₂E₂ is produced.

Fig. R17. 3D soliton molecular assembling with greater complexity. **a.** Temporal evolution calculated from the scalar superposition of the first 10 LP modes. **b.** Three different 3D soliton molecules, i.e., E_2 , CF_2E_2 , and $C_2F_2E_2$ shown as isosurface plots. **c.** Energy distribution of the multimode elements at roundtrips 50×20 and 1470×20 .

Parameters affecting the state of 3D soliton molecules

Here, we want to stress that the configuration of the intermediates, e.g., E_2 and CF_2E_2 , are susceptible to the initial modal condition (i.e., 3D initial condition). This reminds us of the versatile spectral-temporal evolution of single-passing the parabolic multimode optical fiber by controlling the input spatial profile [*Opt. Express* 25, 9078 (2017)].

To address the reviewer's concern, we have provided a corresponding discussion in the main text of the revised manuscript: **Particularly, by correlating the real-time observations to the chemical concepts from an analogous perspective (Supplementary Figure 23), the understandings of these interesting views can be more profound (Supplementary Note 7), e.g., the 3D dual-soliton molecules involving distinguishing transverse modes in analogy with heteronuclear diatomic molecules, and these**

composed of degenerate transverse modes in analogy with homonuclear diatomic molecules (Supplementary Figure 24); furthermore, their hybrid can even give rise to more complicated evolution (Supplementary Figure 25). We also want to stress that, on one hand, the analogous interpretations of the real-time observations of 3D soliton molecules from a chemical perspective can facilitate the understanding of their behaviors and thus dissect the underlying mechanisms; on the other hand, such a platform capable of generating 3D soliton molecules and real-time multi-dimensional observation can be a powerful tool for studying complicated chemical problems, and even various multi-dimensional nonlinear problems in the fields of thermodynamics, hydrodynamics, Bose-Einstein condensates, breathers, rogue waves, etc.

A new section has been added to the Supplementary Information, i.e., **Supplementary Note 7: Underlying dynamic rules of these observations and profound understandings of these interesting views of 3D soliton dynamics**, wherein related discussion has been provided: **Here, we address that the assemblings of dual-soliton or triple-soliton molecules are usually susceptible to the initial modal condition (i.e., 3D initial condition). This reminds us of the versatile spectral-temporal evolution of single-passing the parabolic multimode optical fiber by controlling the input spatial profile¹².**

More discussions have also been provided in the responses to **Comments of Reviewer #2**.

Comment 5: *There are too many abbreviations in the manuscript, and some are difficult to follow. For example, SG in some other papers are normally used as Signal Generation, but here as Speckle Grain. RT can be Room Temperature, but here as Round Trip.*

Authors' response: We appreciate the reviewer for his/her careful reading of our manuscript. We have correspondingly replaced most of the abbreviations with full spellings, except in some specific cases listed in **Table R3**.

Table R3 Abbreviations reserved for specific cases

'RT' reserved in the manuscript	Fig. 3, Fig. 4, Fig. 7
'SG' reserved in the manuscript	SG ₁ , SG ₂ , SG ₃ used in the manuscript
'RT' reserved in the SI	Supplementary Figure 20
'SG' reserved in the SI	SG ₁ , SG ₂ used in the SI

General comment: *Overall, I believe the manuscript is novel, and can be published under revision..*

Authors' response: We appreciate the reviewer for the positive comment.

We hope that these revisions could satisfy the reviewer's concerns and that they meet the publication requirements. Thank you very much for your attention and consideration to our paper.

REVIEWERS' COMMENTS

Reviewer #1 (Remarks to the Author):

The authors have well addressed all my concerns. I would like to suggest acceptance as it is.

Reviewer #2 (Remarks to the Author):

The manuscript has addressed all the comments and can be accepted for publication.

Reviewer #3 (Remarks to the Author):

The author has done a good work to address my comments. Now I support that publication of this work. I believe this work can stimulate my interesting works in nonlinear optics and ultra-fast sensing.

Manuscript #NCOMMS-22-40453A: “Unveiling the complexity of spatiotemporal soliton molecules in real time”

We thank the editor for handling the review process and the reviewers for their constructive comments.

Authors’ point-by-point responses to reviewers’ comments

Reviewer #1: *The authors have well addressed all my concerns. I would like to suggest acceptance as it is.*

Authors’ response: We appreciate the reviewer's insightful comments on our manuscript in the first round of review.

Reviewer #2: *The manuscript has addressed all the comments and can be accepted for publication.*

Authors’ response: We are grateful to the reviewer for his/her insightful comments on our manuscript in the first round of review.

Reviewer #3: *The author has done a good work to address my comments. Now I support that publication of this work. I believe this work can stimulate my interesting works in nonlinear optics and ultra-fast sensing.*

Authors’ response: We thank the reviewer for the positive comment, and the insightful comments on improving our manuscript in the first round of review.